# The Interplay between Distribution Parameters and the Accuracy-Robustness Tradeoff in Classification

**Alireza Mousavi Hosseini** [* 1]   **Amir Mohammad Abouei** [* 1]   **Mohammad Hossein Rohban** [1]

## Abstract

Adversarial training tends to result in models that are less accurate on natural (unperturbed) examples compared to standard models. This can be attributed to either an algorithmic shortcoming or a fundamental property of the training data distribution, which admits different solutions for optimal standard and adversarial classifiers. In this work, we focus on the latter case under a binary Gaussian mixture classification problem. Unlike earlier work, we aim to derive the natural accuracy gap between the optimal Bayes and adversarial classifiers, and study the effect of different distributional parameters, namely separation between class centroids, class proportions, and the covariance matrix, on the derived gap. We show that under certain conditions, the natural error of the optimal adversarial classifier, as well as the gap, are locally minimized when classes are balanced, contradicting the performance of the Bayes classifier where perfect balance induces the worst accuracy. Moreover, we show that with an $\ell_\infty$ bounded perturbation and an adversarial budget of $\epsilon$, this gap is $\Theta(\epsilon^2)$ for the worst-case parameters, which for suitably small $\epsilon$ indicates the theoretical possibility of achieving robust classifiers with near-perfect accuracy, which is rarely reflected in practical algorithms.

## 1. Introduction

It is well known that despite the huge success of deep learning in different tasks such as computer vision and natural language processing, deep models suffer from the existence of adversarial examples (Szegedy et al., 2014). Adversarial training and its variants (Madry et al., 2018; Zhang et al.,

2019; Wang et al., 2020; Carmon et al., 2019) have been one of the most successful defenses against adversarial attacks. However, models obtained via these training methods demonstrate an undesirable drop in the natural accuracy.

There have been many attempts to justify this accuracy drop. Tsipras et al. (2019) showed that the objectives of standard and adversarial training can be fundamentally at odds with each other by proving the existence of plausible settings in which optimal standard and adversarial classifiers exploit inherently different features. Built upon this idea, Zhang et al. (2019) provided an algorithm to train models with a balance between robustness and accuracy. Furthermore, Javanmard et al. (2020) proved the existence of such an innate tradeoff even in the simple case of Gaussian linear regression. Strikingly, Yang et al. (2020) empirically revealed that different classes in benchmark image classification problems are separated enough, which leads to the existence of classifiers that are both Bayes and adversarially optimal. Thus in these cases, the tradeoff can not be attributed to distributional properties.

Another line of work points at the statistical hardness of adversarial training as the cause of the natural accuracy drop. This hardness was first proved by Schmidt et al. (2018) in terms of a higher sample complexity for adversarial learning in a binary Gaussian mixture setting. Bhattacharjee et al. (2020) also showed a higher sample complexity for adversarial training in separated settings. Rice et al. (2020) demonstrated that virtually all adversarial training methods suffer from a huge generalization gap between training and test errors. Empirically, it has been observed that adversarially trained models can enjoy better generalization through a larger training set (Carmon et al., 2019; Raghunathan et al., 2020).

Perhaps closest to this work are that of Dobriban et al. (2020) and Javanmard & Soltanolkotabi (2020). Javanmard & Soltanolkotabi (2020) study the output of the adversarial training algorithm on a similar binary Gaussian mixture problem in high-dimensional settings. However, our work focuses on information-theoretically optimal classifiers, i.e. optimal classifiers when we are aware of the underlying data distribution and have access to its parameters. Dobriban et al. (2020) also study information-theoretically

---

*Equal contribution   [1]Department of Computer Engineering, Sharif University of Technology, Tehran, Iran. Correspondence to: Alireza Mousavi Hosseini <mousavihosseini@ce.sharif.edu>, Amir Mohammad Abouei <amabouei@ce.sharif.edu>.

*Accepted by the ICML 2021 workshop on A Blessing in Disguise: The Prospects and Perils of Adversarial Machine Learning.* Copyright 2021 by the author(s).

optimal standard and adversarial classifiers, and arrive at conditions that imply their difference and subsequently lead to robustness-accuracy tradeoff. However, they do not quantify the exact drop in accuracy and how it can be affected by distributional parameters. We indeed show that even though the two optimal classifiers tend to be different most of the time, with an $\ell_\infty$ adversarial budget of $\epsilon$, their gap in accuracy is $\Theta(\epsilon^2)$ at worst, which can be negligible under practical settings. Besides, we prove the existence of non-trivial global maxima in the natural risk of the optimal adversarial classifier, as well as its gap with the Bayes optimal classifier, under class imbalance.

## 2. Preliminaries and Notation

The training set is defined as $S_n = \{(x^{(1)}, y^{(1)}), ..., (x^{(n)}, y^{(n)})\}$, where $(x^{(i)}, y^{(i)})$ are i.i.d. samples drawn from a distribution $\mathcal{D}$, with $x \in \mathcal{X} \subseteq \mathbb{R}^d$ and $y \in \{-1, +1\}$. For a positive-definite matrix $A$, we define $\|x\|_A = \sqrt{x^T A x}$. Finally, we denote the CDF of the standard Gaussian distribution by $\Phi$.

We define the adversarial risk and the empirical adversarial risk of a classifier with the weight vector $w$ with respect to the loss function $\ell$ over distribution $\mathcal{D}$ as

$$R_{\mathcal{D},\ell}^{adv}(w) = \mathbb{E}_{x,y \sim \mathcal{D}} \left[ \max_{\|x'-x\|_\infty \leq \epsilon} \ell(x', y; w) \right], \quad (1)$$

$$R_{S_n,\ell}^{adv}(w) = \frac{1}{n} \sum_{i=1}^{n} \max_{\|x'^{(i)}-x^{(i)}\|_\infty \leq \epsilon} \ell(x'^{(i)}, y^{(i)}; w). \quad (2)$$

Furthermore, the optimal adversarial weights, when optimized on the entire distribution and the training set, can be defined as

$$w^{adv} = \arg\min_{w \in \mathcal{W}} R_{\mathcal{D},\ell}^{adv}(w), \quad (3)$$

$$w_n^{adv} = \arg\min_{w \in \mathcal{W}} R_{S_n,\ell}^{adv}(w) \quad (4)$$

respectively, where $\mathcal{W}$ is the class of all classifiers (represented by weight vectors), which we search from. We can also define the natural counterpart of the above definitions, denoted by $nat$ superscript rather than $adv$, by setting $\epsilon = 0$.

One quantity that is of particular interest in this paper, is the gap between the natural risk of the optimal adversarial and the Bayes optimal classifiers. Thus, we define

$$G_{\mathcal{D},\ell} = R_{\mathcal{D},\ell}^{nat}(w^{adv}) - R_{\mathcal{D},\ell}^{nat}(w^{nat}), \quad (5)$$

$$G_{\mathcal{D},\ell}^{(n)} = R_{\mathcal{D},\ell}^{nat}(w_n^{adv}) - R_{\mathcal{D},\ell}^{nat}(w_n^{nat}). \quad (6)$$

Although we consider one of the strongest threat models in the literature, the $\ell_\infty$ bounded adversary, our results are easily extensible to other attack norms.

## 3. Warm-up: Linear Loss Function

In this section, we briefly discuss the case of linear classifiers with bounded weights when trained on a linear loss function, i.e. we are interested in the following setting

$$\mathcal{W}_p = \{w \in \mathbb{R}^d \; : \; \|w\|_p \leq W\}, \quad (7)$$

$$\ell_{lin}(x, y; w) = -y\langle w, x \rangle. \quad (8)$$

Considering the simple linear loss function will help us understand how a particular property of the distribution, namely $\mu$, can influence the inherent existence of a tradeoff between natural and adversarial accuracy. The analysis of this section is similar to Chen et al. (2020), which also considered linear loss functions and analyzed the generalization gap between standard and adversarial classifiers. Though they considered only the $\ell_\infty$ regularization case, which does not capture the entire truth, and they restrict their analysis to Gaussian and Bernoulli distributions. The next proposition, proved in Appendix A, will quantify the effect of $\mu$.

**Proposition 1.** *Consider $w^{nat}$ and $w^{adv}$ that are obtained using $\mathcal{W}_p$ and $\ell_{lin}$. For any arbitrary distribution $\mathcal{D}$, let $\mu = \mathbb{E}_{x,y \sim \mathcal{D}}[yx]$. Then, the following conditions are necessary and sufficient to have $w^{nat} = w^{adv}$ for different cases of $p$:*

1. *When $p = 1$, we need $\|\mu\|_\infty \geq \epsilon$ or $\mu = 0$.*

2. *When $p = \infty$, for each $i \in [d]$, we either need $\mu_i = 0$ or $|\mu_i| \geq \epsilon$.*

3. *When $1 < p < \infty$, there must exist $c \geq \epsilon$ such that for each $i \in [d]$, we either have $\mu_i = 0$ or $|\mu_i| = c$.*

While Proposition 1 generally stipulates a large enough distance between class centroids, in the case of $1 < p < \infty$ the condition is much more exclusive, essentially requiring all non-zero elements of $\mu$ to be equal in magnitude. As will be seen later, this condition reappears with the 0-1 loss and Gaussian classification, which hints at its possible importance beyond the settings considered here.

It is easy to provide a finite-sample analysis and observe that the same conditions of Proposition 1 control whether $w_n^{nat}$ and $w_n^{adv}$ converge given enough samples, or that their gap is bounded away from zero. We will formally state and prove this result in Appendix B.

## 4. Complete Analysis of the Tradeoff for Binary Gaussian Mixtures

In this section, we study optimal standard and adversarial classifiers, when samples are generated from a binary Gaussian mixture model. Note that in this section, we choose the 0-1 loss function, which for a linear classifier with weight vector $w$ and bias $w_0$ can be defined as $\ell_{0\text{-}1}(x, y; w) = \mathbb{1}[y(\langle w, x \rangle + w_0) \leq 0]$.

**Definition 2.** Let $\mathcal{D}$ be a distribution on $\mathbb{R}^d \times \mathcal{Y}$, where $\mathcal{Y} = \{-1, +1\}$, such that

$$\mathbb{P}[y = \pm 1] = \pi_{\pm}, \qquad x|y \sim \mathcal{N}(y\mu, \Sigma), \qquad (9)$$

where $p$ is a probability density function. We call $\mathcal{D}$ a $(\mu, \Sigma, \pi_+)$-binary Gaussian mixture.

This class of distributions has been used in many recent works on the theory of adversarial machine learning (Schmidt et al., 2018; Chen et al., 2020; Dobriban et al., 2020), although the form that we consider here is more general than prior works. It is well known that the standard (Bayes) optimal classifier for this distribution, i.e. the classifier that minimizes the natural risk, is a linear classifier with $w = \Sigma^{-1}\mu$ and $w_0 = \frac{\ln(\pi_+/\pi_-)}{2}$. Several prior work have established optimal adversarial classifiers for the binary Gaussian mixture (Bhagoji et al., 2019; Dan et al., 2020; Dobriban et al., 2020). Here, we restate a formulation from Dan et al. (2020), which we found useful for our analysis, with a slight extension to incorporate class imbalance. The proof is deferred to Appendix C.

**Proposition 3.** *Let distribution $\mathcal{D}$ be a $(\mu, \Sigma, \pi_+)$-binary Gaussian mixture according to Definition 2. Then, the optimal adversarial classifier, which is also linear, is given by*

$$z^* = \underset{\|z\|_\infty \leq 1}{\arg\min} \|\mu - \epsilon z\|_{\Sigma^{-1}}^2, \qquad (10)$$

$$w^{adv} = \Sigma^{-1}(\mu - \epsilon z^*), \qquad (11)$$

$$w_0^{adv} = \frac{\ln(\pi_+/\pi_-)}{2}. \qquad (12)$$

For the next sections, we implicitly assume $\epsilon < \|\mu\|_\infty$, omitting the trivial case of $w^{adv} = 0$ according to Proposition 3.

### 4.1. Class Imbalance

In this section, we investigate the effect of class imbalance on the natural accuracy of the optimal adversarial classifier and its gap with the accuracy of the Bayes classifier. Although Dobriban et al. (2020) derived the optimal adversarial classifier for binary and ternary Gaussian mixtures with respect to the $\ell_2$ perturbation norm with imbalanced class priors, they did not analyze the impact of this imbalance on the natural accuracy of that classifier.

It is well known that under constant $\Sigma$ and $\mu$, and varying $\pi_+$, the Bayes optimal classifier for the binary Gaussian mixture model of Definition 2 achieves better accuracies as $\pi_+$ moves away from $\frac{1}{2}$. Perhaps surprisingly, we prove that under certain conditions, this is not the case for the natural risk of the optimal adversarial classifier. The behavior of this risk as a function of $\pi_+$ can have two distinct regimes. In the

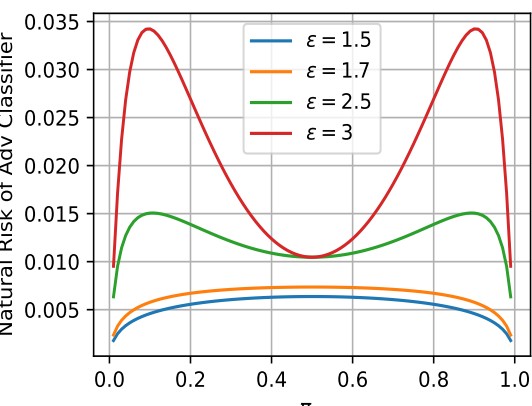

*Figure 1.* The natural risk of the optimal adversarial classifier for different values of $\pi_+$ with fixed parameters are $\mu = (1.5, 2, 4)$ and $\Sigma = 3I_3$. We plot 4 curves according to 4 adversarial budgets in $\ell_\infty$ norm. This figure shows that in a certain regime the natural risk of the optimal adversarial classifier can get worse even as the distribution gets more imbalanced.

first one, it behaves similarly to the natural risk of the Bayes classifier, increasing monotonically before and decreasing monotonically after $\frac{1}{2}$. We call this regime *standard*. In the second regime however, this risk will have 3 extreme values, a local minimum at $\frac{1}{2}$ and two maxima located symmetrically around $\frac{1}{2}$. We call this regime *surprising*. Moreover, this pattern can be seen in the gap between the accuracies of the Bayes and the optimal adversarial classifier as well. In other words, unlike standard classification, when we are in the surprising regime, having perfectly balanced classes makes adversarial classification more naturally accurate and closes its gap with natural classification, compared to a slight class imbalance. This behavior is illustrated in Figure 1. The next theorem, proved in Appendix D, precisely characterizes the condition that separates the two regimes for both the natural risk of the adversarial classifier and the gap.

**Theorem 4.** *Let distribution $\mathcal{D}$ be a $(\mu, \Sigma, \pi_+)$-binary Gaussian mixture according to Definition 2, and let $c := 2\langle\mu, \Sigma^{-1}(\mu - \epsilon z^*)\rangle$ and $d := 2\|\mu - \epsilon z^*\|_{\Sigma^{-1}}$ where $z^*$ is the solution of Equation (10). Consider $R_{\mathcal{D}, \ell_{0-1}}^{nat}(w^{adv})$ as a function of $\pi_+ \in (0, 1)$ under fixed $\Sigma$ and $\mu$. This function is surprising if and only if*

$$c > d^2 \qquad (13)$$

*and standard otherwise. Furthermore, consider $G_{\mathcal{D}, \ell_{0-1}}$, i.e. the gap, as a function of $\pi_+ \in (0, 1)$ under fixed $\Sigma$ and $\mu$. This function is surprising if and only if*

$$\frac{2(\frac{c}{d^2} - 1)e^{-\frac{(\frac{c}{d})^2}{2}}}{d} > -\frac{e^{-\frac{\|\mu\|_{\Sigma^{-1}}^2}{2}}}{2\|\mu\|_{\Sigma^{-1}}} \qquad (14)$$

*and standard otherwise.*

*Remark.* As expected, the satisfaction of (13) also implies the satisfaction of (14).

Although not proved formally, we believe that the transition to the surprising regime generally occurs with a non-trivially (less than $\|\mu\|_\infty$) large enough $\epsilon$. For instance, let $\Sigma$ be diagonal and let all elements of $\mu$ have the same magnitude $|\mu|$. Then, $\epsilon > \frac{|\mu|}{2}$ would suffice to satisfy (13) and subsequently (14). Note that in this case, the latter is satisfied for all $\epsilon$.

### 4.2. Gap Lower and Upper Bounds

In this section, we focus on the case where $\pi_+ = \pi_-$. We begin our discussion by specifying the conditions under which the adversarial and Bayes optimal classifiers are equivalent, i.e. $w^{nat}$ and $w^{adv}$ are parallel, in the following proposition which is proved in Appendix E. This is indeed an extension to the results of Dobriban et al. (2020), where they only consider isotropic covariance matrices.

**Theorem 5.** *Let distribution $\mathcal{D}$ be a $(\mu, \Sigma, \frac{1}{2})$-binary Gaussian mixture according to Definition 2. Then the optimal adversarial classifier is also a Bayes optimal classifier if and only if a constant $c \geq \epsilon$ exists such that*

$$\forall i \in [d] \quad (\Sigma^{-1}\mu)_i = 0 \implies |\mu_i| \leq c, \tag{15}$$

$$\forall i \in [d] \quad (\Sigma^{-1}\mu)_i \neq 0 \implies \mu_i = c\,\mathrm{sign}((\Sigma^{-1}\mu)_i). \tag{16}$$

*Remark.* For a diagonal $\Sigma$, the condition in Theorem 5 will be the same as the condition for $1 < p < \infty$ in Proposition 1, which demands a strong symmetry between all non-zero elements of $\mu$ that is unimaginable to hold in practice.

As a consequence of Theorem 5, it is reasonable to assume that in most natural settings, there is indeed a tradeoff between optimizing standard accuracy and robustness. However, the natural question to ask is, when the conditions stated above are not met, how much accuracy drop should we expect? The next theorem, proved in Appendix F, addresses this question.

**Theorem 6.** *Let distribution $\mathcal{D}$ be a $(\mu, \Sigma, \frac{1}{2})$-binary Gaussian mixture according to Definition 2, and let $\epsilon \leq \min(\frac{\|\mu\|_{\Sigma^{-1}}^2}{2\|\Sigma^{-1}\mu\|_1}, \frac{\|\mu\|_{\Sigma^{-1}}}{2\sqrt{C_{\Sigma,\mu}}})$ where $C_{\Sigma,\mu} = \|\mathrm{sign}(\Sigma^{-1}\mu)\|_{\Sigma^{-1}}^2 - \frac{\|\Sigma^{-1}\mu\|_1^2}{\|\mu\|_{\Sigma^{-1}}^2}$. Then we have*

$$G_{\mathcal{D},\ell_{0\text{-}1}} \leq \Phi\left(-\|\mu\|_{\Sigma^{-1}} + \frac{2C_{\Sigma,\mu}\epsilon^2}{\|\mu\|_{\Sigma^{-1}}}\right) - \Phi(-\|\mu\|_{\Sigma^{-1}})$$

$$\leq \frac{2e^{\frac{-\|\mu\|_{\Sigma^{-1}}^2}{8}}C_{\Sigma,\mu}\epsilon^2}{\sqrt{2\pi}\|\mu\|_{\Sigma^{-1}}}. \tag{17}$$

*Moreover, when $\Sigma$ is diagonal and $\min_{i\in[d]} |\mu_i| \geq \epsilon$, we*

*have*

$$G_{\mathcal{D},\ell_{0\text{-}1}} \geq \frac{e^{\frac{-\|\mu\|_{\Sigma^{-1}}^2}{2}}C_{\Sigma,\mu}\epsilon^2}{3\sqrt{2\pi}\|\mu\|_{\Sigma^{-1}}}. \tag{18}$$

*Remark.* One can easily verify $C_{\Sigma,\mu} \geq 0$ using the Hölder's inequality. In fact, when $C_{\Sigma,\mu} = 0$, there will be no tradeoff.

Equation (90) presents a tighter upper bound without the requirement of $\epsilon \leq \frac{\|\mu\|_{\Sigma^{-1}}}{2\sqrt{C_{\Sigma,\mu}}}$, which is omitted here for simpler presentation.

There are two interesting observations to be made surrounding Theorem 6:

- As the optimal standard accuracy tends to one, i.e. $\|\mu\|_{\Sigma^{-1}} \to \infty$, the gap vanishes exponentially quickly.

- In high dimensional cases (as $d$ grows), an interesting setting occurs when $\frac{\sqrt{C_{\Sigma,\mu}}}{\|\mu\|_{\Sigma^{-1}}} \in \mathcal{O}(1)$, e.g. when $\Sigma$ is isotropic and $\mu$ has equal constant elements. In this case, the gap remains negligible with the growth of $d$, especially if $\epsilon \ll \frac{\sqrt{C_{\Sigma,\mu}}}{\|\mu\|_{\Sigma^{-1}}}$.

In order to better understand the information-theoretical limit provided by Theorem 6, we provide further numerical experiments in Appendix G.

## 5. Conclusion

In this work, we studied the effect of the parameters of the underlying generative distribution on the natural accuracy gap between the Bayes optimal and the optimal adversarial classifiers. In particular, when the underlying distribution is a binary Gaussian mixture, we proved an interesting characteristic of adversarial classifiers, that their natural error and its gap with the Bayes classifer are locally minimized when classes are balanced, contrary to the Bayes error rate that is maximized in this case. Furthermore, we specified the exact conditions on $\mu$ and $\Sigma$ when there is no tradeoff. Although these conditions are highly unlikely to be met, we showed that the gap is roughly $\Theta(\epsilon^2)$ in the worst-case scenario, which can be negligible in many applications with small enough $\epsilon$. These results can help as a guideline to ensure that newly developed robust learning algorithms are close to information-theoretical optimality.

While this work addressed the question of information-theoretically optimal achievable gap, it is interesting to study the limits of finite-sample algorithms next, which can provide valuable insights for practitioners in designing robust deep learning algorithms.

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

# Appendices

## A. Proof of Proposition 1

We begin by proving the following lemma, which will be useful in proving Theorem 8 as well.

**Lemma 7.** *Let $w^{adv}$, $w^{nat}$ and $\mu$ be defined as in Proposition 1. Then, for any arbitrary distribution $\mathcal{D}$, we have*

$$
w^{adv} = \begin{cases} W\beta_\epsilon(\mu) & p = 1 \\ W\frac{\eta_{\epsilon,p}(\mu)}{\|\eta_{\epsilon,p}(\mu)\|_p} & 1 < p < \infty \text{ and } \|\mu\|_\infty > \epsilon \\ 0 & 1 < p < \infty \text{ and } \|\mu\|_\infty \leq \epsilon \\ W\gamma_\epsilon(\mu) & p = \infty \end{cases} \tag{19}
$$

*where $\beta_\epsilon, \eta_{\epsilon,p}, \gamma_\epsilon : \mathbb{R}^d \to \mathbb{R}^d$ are defined as*

$$
\beta_\epsilon(\mu)_i = \mathbb{1}(i = \arg\max_{j\in[d]} |\mu_j|)\mathbb{1}(\|\mu\|_\infty \geq \epsilon)\operatorname{sign}(\mu_i), \tag{20}
$$

$$
\eta_{\epsilon,p}(\mu)_i = \mathbb{1}(|\mu_i| \geq \epsilon)|\mu_i - \epsilon\operatorname{sign}(\mu_i)|^{\frac{1}{p-1}}\operatorname{sign}(\mu_i), \tag{21}
$$

$$
\gamma_\epsilon(\mu)_i = \operatorname{sign}(\mu_i)\mathbb{1}(|\mu_i| \geq \epsilon). \tag{22}
$$

*Moreover, $w^{nat}$ can be obtained by setting $\epsilon = 0$.*

First, we simplify the adversarial loss as follows

$$
\ell(x', y, w) = \max_{\|\delta\|_\infty \leq \epsilon} -y\langle w, x + \delta\rangle = -y\langle w, x\rangle + \max_{\|\delta\|_\infty \leq \epsilon} \langle w, -y\delta\rangle = -y\langle w, x\rangle + \epsilon\|w\|_1
$$

where we used the fact that $\ell_1$ and $\ell_\infty$ norms are dual norms for the last equality.

Thus, our problem is simplified to

$$
w^{adv} = \arg\min_{\|w\|_p \leq W} R^{adv}_{\mathcal{D},\ell}(w) = \arg\min_{\|w\|_p \leq W} -\langle w, \mu\rangle + \epsilon\|w\|_1.
$$

We start with the case of $p = 1$. Using the Holder's inequality we have

$$
R^{adv}_{\mathcal{D},\ell}(w) \geq \|w\|_1(\epsilon - \|\mu\|_\infty).
$$

If $\epsilon > \|\mu\|_\infty$, then for any $w \neq 0$, $R^{adv}_{\mathcal{D},\ell}$ will be strictly positive, thus the optimal solution would be $w^{adv} = 0$. Otherwise, with setting $w^{adv} = W\beta_\epsilon(\mu)$ where $\beta_\epsilon$ is defined as in Equation (20), $R^{adv}_{\mathcal{D},\ell}$ will achieve the minimum of its lower bound and is therefore minimized.

Next, we move to the case where $1 < p < \infty$. In this case, we will form the Lagrange function

$$
\mathcal{L}(w, \lambda) = -\langle w, \mu\rangle + \epsilon\|w\|_1 + \lambda(\|w\|_p^p - W^p).
$$

Note that for minimizing the Lagrange function, we can decouple it and minimize the term corresponding to each $w_i$ individually. Thus, we seek to minimize

$$
-w_i\mu_i + \epsilon|w_i| + \lambda|w_i|^p.
$$

Using the above equation's subgradients for $\lambda > 0$, we arrive at the following solution

$$
w_i^{adv} = \frac{\eta_{\epsilon,p}(\mu)_i}{(\lambda p)^{\frac{1}{p-1}}}.
$$

Now, by choosing the appropriate $\lambda$ such that $\|w\|_p = W$ we will arrive at the desired result.

Finally, we present the proof for the $p = \infty$ case. In this case, Equation 4 can be written as

$$w^{adv} = \underset{\|w\|_\infty \leq W}{\arg\min} -\langle w, \mu \rangle + \epsilon \|w\|_1 = \underset{\forall j \in [d] \; |w_j| \leq W}{\arg\min} \sum_{j=1}^{d} -w_j \mu_j + \epsilon |w_j|.$$

Therefore, we can optimize each $w_j$ separately. Note that when $|\mu_j| < \epsilon$ the term corresponding to $w_j$ is non-negative and will be minimized by choosing $w_j = 0$. When $|\mu_j| = \epsilon$ choosing any $w_j$ such that $w_j \mu_j \geq 0$ will be a minimizer, and we choose $w_j = W \operatorname{sign}(\mu_j)$. Otherwise, choosing $w_j = W \operatorname{sign}(\mu_j)$ will minimize the corresponding term, which will finalize the proof.

From here, one can easily verify the truth of Proposition 1 using Equations (20), (21), and (22), by comparing the cases of $\epsilon > 0$ and $\epsilon = 0$ (which gives the standard classifier, i.e. $w^{nat}$). $\qquad \square$

## B. Finite-Sample Analysis of Training with the Linear Loss Function

First, it is worthwhile to note that unlike the rest of the paper, the weights obtained in this section are stochastic as they are learned using the training set $S_n$, without having knowledge about the distribution.

As a brief description, the theorem presented below states that when we perform training over $S_n$ using $\mathcal{W}_p$ and $\ell_{\text{lin}}$ as in (7) and (8), then:

- When $p = 1$, if $\|\mu\|_\infty > \epsilon$ then we have $w_n^{adv} = w_n^{nat}$ w.h.p., and if $0 < \|\mu\|_\infty < \epsilon$ we have $G_{\mathcal{D},\ell_{\text{lin}}}^{(n)} = W\|\mu\|_\infty$ w.h.p.

- When $p = \infty$, if $\min_{i \in [d]} |\mu_i| \geq \epsilon$, then $w_n^{adv} = w_n^{nat}$ w.h.p., and otherwise, i.e. if there exists an index $j$ such that $0 < |\mu_j| < \epsilon$, then $G_{\mathcal{D},\ell_{\text{lin}}}^{(n)} \geq W|\mu_j|$ w.h.p.

Note that these are the same conditions in order to have $w^{nat} = w^{adv}$ in Proposition 1, indicating that the conditions not only characterize when the information-theoretically optimal classifiers are identical, but also when the result of the two training algorithms are *probably* the same.

**Theorem 8.** *Consider $w_n^{nat}$ and $w_n^{adv}$ obtained via training on $S_n$ using $\mathcal{W}_p$ and $\ell_{lin}$ as in Equations (7) and (8). Let $\mathcal{X}$ be such that $\sup_{x \in \mathcal{X}} \|x\|_\infty \leq A < \infty$, and let $\mu = \mathbb{E}_{(x,y) \sim \mathcal{D}}[yx]$. Define $r := \|\mu\|_\infty - \max_{\{k \in [d]: |\mu_k| < \|\mu\|_\infty\}} |\mu_k|$ and let $r = 0$ when all elements of $\mu$ are equal. Define $\tau := \min_{j \in [d]} |\mu_j| - \epsilon$. Then*

1. *When $p = 1$, if $\|\mu\|_\infty \geq \epsilon$ then we have*

$$\mathbb{P}[w_n^{adv} = w_n^{nat}] \geq 1 - 2\exp\left(\frac{-n(\|\mu\|_\infty - \epsilon)^2}{2A^2}\right). \tag{23}$$

   *Furthermore, if $0 < \|\mu\|_\infty < \epsilon$, then we have*

$$\mathbb{P}[R_{\mathcal{D},\ell}^{nat}(w_n^{adv}) - R_{\mathcal{D},\ell}^{nat}(w_n^{nat}) = W\|\mu\|_\infty] \geq 1 - 4d\mathbb{1}[r > 0]\exp\left(\frac{-nr^2}{8A^2}\right) - 2d\exp\left(\frac{-n(\epsilon - \|\mu\|_\infty)^2}{2A^2}\right). \tag{24}$$

2. *When $p = \infty$, if $\min_{i \in [d]} |\mu_i| \geq \epsilon$ we have*

$$\mathbb{P}[w_n^{adv} = w_n^{nat}] \geq 1 - 2d\exp\left(\frac{-n\tau^2}{2A^2}\right). \tag{25}$$

   *Conversely, if there exists an index $j$ such that $0 < |\mu_j| < \epsilon$, then we have*

$$\mathbb{P}[R_{\mathcal{D},\ell}^{nat}(w_n^{adv}) - R_{\mathcal{D},\ell}^{nat}(w_n^{nat}) \geq W|\mu_j|] \geq 1 - 2\exp\left(\frac{-n\min(\epsilon - |\mu_j|, |\mu_j|)^2}{2A^2}\right). \tag{26}$$

*Remark.* $w_n^{adv} = w_n^{nat}$ can be ambiguous when the optimal classifiers are not unique. By this notation, we mean that there is at least some classifier that is both naturally and adversarially optimal. On the other hand, when talking about $R_{\mathcal{D},\ell}^{nat}(w_n^{adv}) - R_{\mathcal{D},\ell}^{nat}(w_n^{nat})$, the statement holds for all possible solutions $w_n^{nat}$ and $w_n^{adv}$.

*Proof.* Let $\hat{\mu} := \frac{1}{n} \sum_{i=1}^{n} y^{(i)} x^{(i)}$ and note that one can simply substitute $\mu$ with $\hat{\mu}$ in Lemma 7 to obtain closed-form expressions for $w_n^{adv}$ and $w_n^{nat}$. First, consider the case where $p = 1$ and $\|\mu\|_\infty > \epsilon$. Choose a fixed $i$ such that $|\mu_i| = \|\mu\|_\infty$. In order to have $w_n^{adv} = w_n^{nat}$, it suffices to have $\|\hat{\mu}\|_\infty \geq \epsilon$, and consequently it suffices that $|\hat{\mu}_i| \geq \epsilon$. Using the Hoeffding inequality, we have

$$\mathbb{P}[|\hat{\mu}_i| \geq \epsilon] \geq \mathbb{P}[|\mu_i - \hat{\mu}_i| \leq |\mu_i| - \epsilon] \geq 1 - 2 \exp\left( \frac{-n(|\mu_i| - \epsilon)^2}{2A^2} \right)$$

proving Equation (23).

Next, in order to prove (24), we show that whenever $0 < \|\mu\|_\infty < \epsilon$, we have $R_{\mathcal{D},\ell}^{nat}(w_n^{adv}) = 0$ and $R_{\mathcal{D},\ell}^{nat}(w_n^{nat}) = -W\|\mu\|_\infty$ *w.h.p.* The first condition will be met whenever $\|\hat{\mu}\|_\infty < \epsilon$, which implies $w_n^{adv} = 0$. For the probability of this event, we have

$$\begin{aligned}
\mathbb{P}[\forall k \quad |\hat{\mu}_k| < \epsilon] &\geq 1 - \sum_{k=1}^{d} \mathbb{P}[|\hat{\mu}_k| \geq \epsilon] \geq 1 - \sum_{k=1}^{d} \mathbb{P}[|\hat{\mu}_k - \mu_k| \geq \epsilon - |\mu_k|] \\
&\geq 1 - \sum_{k=1}^{d} \mathbb{P}[|\hat{\mu}_k - \mu_k| \geq \epsilon - \|\mu\|_\infty] \\
&\geq 1 - 2d \exp\left( \frac{-n(\epsilon - \|\mu\|_\infty)^2}{2A^2} \right).
\end{aligned} \tag{27}$$

The second condition can be achieved whenever for a fixed $i$ with $|\mu_i| = \|\mu\|_\infty$ and for every $k$ such that $|\mu_i| > |\mu_k|$ we have $|\hat{\mu}_i| > |\hat{\mu}_k|$. Using the triangle inequality, it suffices to have $|\hat{\mu}_i - \mu_i| < \frac{|\mu_i| - |\mu_k|}{2}$ and $|\hat{\mu}_k - \mu_k| < \frac{|\mu_i| - |\mu_k|}{2}$.

$$\begin{aligned}
\mathbb{P}[\forall k \; |\mu_k| < |\mu_i| \implies |\hat{\mu}_k| < |\hat{\mu}_i|] &\geq 1 - \sum_{\{k:|\mu_k| < |\mu_i|\}} \mathbb{P}[|\hat{\mu}_k| \geq |\hat{\mu}_i|] \\
&\geq 1 - \sum_{\{k:|\mu_k| < |\mu_i|\}} \mathbb{P}\left[ \left\{ |\hat{\mu}_i - \mu_i| \geq \frac{|\mu_i| - |\mu_k|}{2} \right\} \cup \left\{ |\hat{\mu}_k - \mu_k| \geq \frac{|\mu_i| - |\mu_k|}{2} \right\} \right] \\
&\geq 1 - \sum_{\{k:|\mu_k| < |\mu_i|\}} \mathbb{P}\left[ \left\{ |\hat{\mu}_i - \mu_i| \geq \frac{r}{2} \right\} \cup \left\{ |\hat{\mu}_k - \mu_k| \geq \frac{r}{2} \right\} \right] \\
&\geq 1 - 4d\mathbb{1}[r > 0] \exp\left( \frac{-nr^2}{8A^2} \right).
\end{aligned} \tag{28}$$

Combining (27) and (28) completes the proof of (24).

Next, we consider the $p = \infty$ and $\min_{i \in [d]} |\mu_i| \geq \|\mu\|_\infty$ case. For $w_n^{adv}$ and $w_n^{nat}$ to be different we need $|\hat{\mu}_j| < \epsilon$ to hold for at least one $j \in [d]$. Thus, using the union bound we have

$$\begin{aligned}
\mathbb{P}[w_n^{adv} \neq w_n^{nat}] &\leq \sum_{j=1}^{d} \mathbb{P}[|\hat{\mu}_j| < \epsilon] \\
&\leq \sum_{j=1}^{d} \mathbb{P}[|\mu_j - \hat{\mu}_j| \geq |\mu_j| - \epsilon] \\
&\leq 2 \sum_{j=1}^{d} \exp\left( \frac{-n(|\mu_j| - \epsilon)^2}{2A^2} \right) \leq 2d \exp\left( \frac{-n\tau^2}{2A^2} \right).
\end{aligned}$$

which proves (25).

Finally, for the $p = \infty$ and $0 < |\hat{\mu}_j| < \epsilon$ case, we need to show that $0 < |\hat{\mu}_j| < \epsilon$ *w.h.p.* As a result, the corresponding element in $w_n^{nat}$ will be nonzero, while the corresponding element in $w_n^{adv}$ will be zero, which will cause the difference in natural risks. For $0 < |\hat{\mu}_j| < \epsilon$ it is sufficient to have $|\hat{\mu}_j - \mu_j| < \min(\epsilon - |\mu_j|, |\mu_j|)$. Thus

$$\mathbb{P}[0 < |\hat{\mu}_j| < \epsilon] \geq \mathbb{P}[|\hat{\mu}_j - \mu_j| < \min(\epsilon - |\mu_j|, |\mu_j|)] \geq 1 - 2\exp\left(\frac{-n\min(\epsilon - |\mu_j|, |\mu_j|)^2}{2A^2}\right). \tag{29}$$

With this, the proof of (26) is completed and all cases have been covered. $\square$

## C. Proof of Proposition 3

Several proofs of this statement already exist in the literature (Dan et al., 2020; Bhagoji et al., 2019; Dobriban et al., 2020). Here, we adopt a proof due to Dan et al. (2020) with a slight modification to incorporate class imbalance.

We begin by finding the optimal linear adversarial classifier, and then show that no other (non-linear) classifier can achieve better adversarial accuracy. First, we derive a closed form expression for the adversarial risk of a linear classifier with weight vector $w$ and bias $w_0$. For a sample $(x, y)$, the adversary will introduce a noise $\delta$ such that $y\langle w, x + \delta\rangle = \min_{\|v\|_\infty \leq \epsilon} y\langle w, x + v\rangle = y\langle w, x\rangle - \epsilon\|w\|_1$. Therefore

$$R_{\mathcal{D},\ell}^{adv}(w, w_0) = \pi_+ \, \mathbb{P}_{x \sim \mathcal{N}(\mu, \Sigma)}[\langle w, x\rangle - \epsilon\|w\|_1 + w_0 \leq 0] + \pi_- \, \mathbb{P}_{x \sim \mathcal{N}(-\mu, \Sigma)}[\langle w, x\rangle + \epsilon\|w\|_1 + w_0 \geq 0]$$

$$= \pi_+ \Phi\left(\frac{-\langle w, \mu\rangle + \epsilon\|w\|_1 - w_0}{\|w\|_\Sigma}\right) + \pi_- \Phi\left(\frac{-\langle w, \mu\rangle + \epsilon\|w\|_1 + w_0}{\|w\|_\Sigma}\right) \tag{30}$$

Since multiplying $w$ and $w_0$ by any positive scalar does not change the risk, there exist optimal solutions for which $\|w\|_\Sigma = 1$. Applying this restriction, we end up with

$$R_{\mathcal{D},\ell}^{adv}(w, w_0) = \pi_+ \Phi(-\langle w, \mu\rangle + \epsilon\|w\|_1 - w_0) + \pi_- \Phi(-\langle w, \mu\rangle + \epsilon\|w\|_1 + w_0). \tag{31}$$

It is easy to see that optimizing $w$ is independent from $w_0$, and we should minimize $-\langle w, \mu\rangle + \epsilon\|w\|_1$ subject to $\|w\|_\Sigma = 1$. We can convexify the problem by relaxing the condition to $\|w\|_\Sigma \leq 1$ and verifying that the solution meets $\|w\|_\Sigma = 1$ at the end. Now, we can restate the problem of $\min_{\|w\|_\Sigma \leq 1} -\langle w, \mu\rangle + \epsilon\|w\|_1$ as $\min_{\|w\|_\Sigma \leq 1} \max_{\|z\|_\infty \leq 1} -\langle w, \mu - \epsilon z\rangle$. Since the objective is concave-convex and the constraints are convex sets, we can change the min-max order using von Neumann's Minimax theorem, which results in $\max_{\|z\|_\infty \leq 1} \min_{\|w\|_\Sigma \leq 1} -\langle w, \mu - \epsilon z\rangle$. It is straightforward to see that the solution to the inner maximization is given by $w^{adv} = \frac{\Sigma^1(\mu - \epsilon z)}{\|\mu - \epsilon\|_{\Sigma^{-1}}}$, and we are left with the outer maximization problem which is $\max_{\|z\|_\infty \leq 1} -\|\mu - \epsilon z\|_{\Sigma^{-1}}$.

We further show that we must have $z_i^* = \text{sign}(w_i^{adv})$ whenever $w_i^{adv} \neq 0$, which will be used later in the proof. Returning to the problem of $z^* = \arg\min_{\|z\|_\infty \leq 1} \|\mu - \epsilon z\|_{\Sigma^{-1}}^2$ and using the method of Lagrange multipliers with dual variables $u_1, ..., u_d$, we have

$$\mathcal{L}(z, u) = \langle \mu - \epsilon z, \Sigma^{-1}(\mu - \epsilon z)\rangle + \sum_{i=1}^{d} u_i(|z_i| - 1). \tag{32}$$

At the optimal point, the subgradient satisfies $\partial_z \mathcal{L}(z^*, u^*) = 0$. Therefore,

$$-2(\Sigma^{-1}(\mu - \epsilon z^*))_i + u_i^* \partial_{z_i}|z_i^*| = 0 \implies -2\|\mu - \epsilon z^*\|_{\Sigma^{-1}} w_i^{adv} + u_i^* \partial_{z_i}|z_i^*| = 0 \tag{33}$$

where $\partial_{z_i}|z_i^*| = \text{sign}(z_i^*)$ whenever $z_i^* \neq 0$ and any real number in $[-1, 1]$ otherwise. One can see that when $w_i^{adv} \neq 0$, we must have $u_i^* > 0$ and $z_i^* = \text{sign}(w_i^{adv})$ due to the complementary slackness condition.

It remains to find the optimal value for $w_0$. Let $K := \langle w^{adv}, \mu\rangle - \epsilon\|w^{adv}\|_1$. Then

$$\frac{\partial R_{\mathcal{D},\ell}^{adv}(w^{adv}, w_0)}{\partial w_0} = \frac{e^{\frac{-K^2 - w_0^2}{2}}}{\sqrt{2\pi}}\left(-e^{\ln(\pi_+) - Kw_0} + e^{\ln(\pi_-) + Kw_0}\right) \tag{34}$$

Setting $\frac{\partial R_{\mathcal{D},\ell}^{adv}(w^{adv}, w_0)}{\partial w_0} = 0$ leads to $w_0^* = \frac{\ln(\pi_+/\pi_-)}{2K}$, and one can observe that $\frac{\partial R_{\mathcal{D},\ell}^{adv}(w^{adv}, w_0)}{\partial w_0} > 0$ if and only if $w_0 > w_0^*$, thus $w_0^*$ is achieving the minimum value of $R_{\mathcal{D},\ell}^{adv}(w^{adv}, w_0)$. Furthermore, by plugging $w^{adv}$ in $K$, we have

$K = \|\mu - \epsilon z\|_{\Sigma^{-1}}$. Since multiplying both $w^{adv}$ and $w_0^*$ by a positive scalar will not change the classifier, we multiply both by $K$, thus arriving at $w^{adv} = \Sigma^{-1}(\mu - \epsilon z^*)$ and $w_0^{adv} = \frac{\ln(\pi_+/\pi_-)}{2}$, as the theorem claims.

It remains to show that this linear classifier is indeed optimal among all (potentially non-linear) classifiers. We first present a slightly extended version of Lemma 6.4 from Dan et al. (2020), and apply it in a similar fashion to their work.

**Lemma 9.** *Let distribution $\mathcal{D}$ and $\mathcal{D}'$ be a $(\mu, \Sigma, \pi_+)$ and a $(\mu - \epsilon z^*, \Sigma, \pi_+)$-binary Gaussian mixture respectively. For any classifier $f : \mathbb{R}^d \to \{-1, +1\}$ we have*

$$R_{\mathcal{D}, \ell_{0\text{-}1}}^{adv}(f) \geq R_{\mathcal{D}', \ell_{0\text{-}1}}^{nat}(f) \geq \pi_+ \Phi\left(-\|w^{adv}\|_\Sigma - \frac{w_0^{adv}}{\|w^{adv}\|_\Sigma}\right) + \pi_- \Phi\left(-\|w^{adv}\|_\Sigma + \frac{w_0^{adv}}{\|w^{adv}\|_\Sigma}\right). \tag{35}$$

*Proof.* First, note that we have

$$R_{\mathcal{D}, \ell_{0\text{-}1}}^{adv}(f) = \pi_+ \mathbb{E}_{x \sim \mathcal{N}(\mu, \Sigma)}\left[\max_{\{x' \,:\, \|x'-x\|_\infty \leq \epsilon\}} \mathbb{1}(f(x') \leq 0)\right] + \pi_- \mathbb{E}_{x \sim \mathcal{N}(-\mu, \Sigma)}\left[\max_{\{x' \,:\, \|x'-x\|_\infty \leq \epsilon\}} \mathbb{1}(f(x') \geq 0)\right]. \tag{36}$$

Moreover, by definition we have $\|\epsilon z^*\|_\infty \leq \epsilon$, which implies that

$$\mathbb{E}_{x \sim \mathcal{N}(\mu, \Sigma)}\left[\max_{\{x' \,:\, \|x'-x\|_\infty \leq \epsilon\}} \mathbb{1}(f(x') \leq 0)\right] \geq \mathbb{E}_{x \sim \mathcal{N}(\mu, \Sigma)}[\mathbb{1}(f(x - \epsilon z^*) \leq 0)] = \mathbb{E}_{x \sim \mathcal{N}(\mu - \epsilon z^*, \Sigma)}[\mathbb{1}(f(x) \leq 0)] \tag{37}$$

and a similar lower bound for the negative case. Thus we have

$$R_{\mathcal{D}, \ell_{0\text{-}1}}^{adv}(f) \geq \pi_+ \mathbb{E}_{x \sim \mathcal{N}(\mu - \epsilon z^*, \Sigma)}[\mathbb{1}(f(x) \leq 0)] + \pi_- \mathbb{E}_{x \sim \mathcal{N}(-\mu + \epsilon z^*, \Sigma)}[\mathbb{1}(f(x) \geq 0)] = R_{\mathcal{D}', \ell_{0\text{-}1}}^{nat}(f) \tag{38}$$

which completes the proof of the first inequality. For the second inequality, note that the Bayes optimal classifier for $\mathcal{D}'$ is indeed a linear classifier with $w^{adv} = \Sigma^{-1}(\mu - \epsilon z^*)$ and $w_0^{adv} = \frac{\ln(\pi_+/\pi_-)}{2}$, and any other classifier has a higher natural risk compared to the stated Bayes risk. $\square$

Now, all we need to show is that the adversarial risk of $(w^{adv}, w_0^{adv})$ is equal to the lower bound stated in Lemma 9. For this, note that whenever $w_i^{adv} \neq 0$ we have $z_i^* = \text{sign}(w_i^{adv})$. As a result, we have $\langle z^*, w^{adv} \rangle = \|w^{adv}\|_1$ and thus $\langle w^{adv}, \mu \rangle - \epsilon \|w^{adv}\|_1 = \langle w^{adv}, \mu - \epsilon z^* \rangle$. Plugging this result into Equation (30) implies that the lower bound in Lemma 9 is equal to $R_{\mathcal{D}, \ell_{0\text{-}1}}^{adv}(w^{adv}, w_0^{adv})$ which demonstrates that for any classifier $f$ we have $R_{\mathcal{D}, \ell_{0\text{-}1}}^{adv}(f) \geq R_{\mathcal{D}, \ell_{0\text{-}1}}^{adv}(w^{adv}, w_0^{adv})$. With that, the proof of Proposition 3 is complete. $\square$

## D. Proof of Theorem 4

Both conditions in Theorem 4 have a similar proof. First, for the sake of simplicity, we define

$$c := 2\langle \mu, \Sigma^{-1}(\mu - \epsilon z^*) \rangle, \tag{39}$$

$$d := 2\|\mu - \epsilon z^*\|_{\Sigma^{-1}} \tag{40}$$

and $\pi := \pi_+$. The proof is based on analyzing the second derivative of our desired function with respect to $\pi$. Define the following function which simplifies the proof

$$F(\pi) = F_{c,d}(\pi) := \pi \Phi\left(\frac{-c - \ln \frac{\pi}{1-\pi}}{d}\right). \tag{41}$$

From Proposition 3 and Equation (30) by setting $\epsilon = 0$, it is obvious that

$$R_{c,d}(\pi) := R_{\mathcal{D}, \ell}^{nat}(w^{adv}, w_0^{adv}) = R_{\mathcal{D}, \ell}^{nat}\left(w^{adv}, \frac{1}{2} \ln \frac{\pi}{1-\pi}\right) = F(\pi) + F(1 - \pi). \tag{42}$$

The next lemmas are essential for proving both theorems.

**Lemma 10.** *Consider $c$ and $d$ defined by Equations (39) and (40). We have*

$$\frac{c}{d^2} \geq \frac{1}{2} \tag{43}$$

*Proof.* Replace c,d by their values. we have

$$\frac{c}{d^2} = \frac{\langle \mu, \Sigma^{-1}(\mu - \epsilon z^*)\rangle}{2\|\mu - \epsilon z^*\|_{\Sigma^{-1}}^2} \geq \frac{1}{2}$$

$$\iff \langle \mu, \Sigma^{-1}(\mu - \epsilon z^*)\rangle \geq \langle \mu - \epsilon z^*, \Sigma^{-1}(\mu - \epsilon z^*)\rangle$$

$$\iff \langle z^*, \Sigma^{-1}\mu\rangle \geq \epsilon\langle z^*, \Sigma^{-1}z^*\rangle$$

$$\iff \langle z^*, \Sigma^{-1}(\mu - \epsilon z^*)\rangle \geq 0$$

Recall from Appendix C that we have $z_i^* = \text{sign}(\Sigma^{-1}(\mu - \epsilon z^*))_i$ whenever $(\Sigma^{-1}(\mu - \epsilon z^*))_i \neq 0$, hence the last inequality holds. Note that if $\epsilon = 0$, we will have $\frac{c}{d^2} = \frac{1}{2}$. □

**Lemma 11.** *Consider the function $F$ defined by Equation* (41). *The second derivative of this function is*

$$F''(\pi) = \frac{\phi\left(\frac{-c - \ln\frac{\pi}{1-\pi}}{d}\right)}{d\pi(1-\pi)^2}\left(\frac{1}{d^2}(c + \ln\frac{\pi}{1-\pi}) - 1\right) \tag{44}$$

*where $\phi(x) = \frac{d}{dx}\Phi(x) = \frac{1}{\sqrt{2\pi}}e^{-\frac{x^2}{2}}$.*

*Proof.* First, we have

$$F'(\pi) = \frac{d}{d\pi}\pi\Phi\left(\frac{-c - \ln\frac{\pi}{1-\pi}}{d}\right) = \Phi\left(\frac{-c - \ln\frac{\pi}{1-\pi}}{d}\right) - \frac{1}{d(1-\pi)}\phi\left(\frac{-c - \ln\frac{\pi}{1-\pi}}{d}\right). \tag{45}$$

Now, differentiating $F'(\pi)$ results in

$$F''(\pi) = \frac{d}{d\pi}F'(\pi) = -\frac{1}{d\pi(1-\pi)}\phi\left(\frac{-c - \ln\frac{\pi}{1-\pi}}{d}\right) - \frac{1}{d(1-\pi)^2}\phi\left(\frac{-c - \ln\frac{\pi}{1-\pi}}{d}\right)$$

$$- \frac{1}{d^2\pi(1-\pi)^2}\left(\frac{-c - \ln\frac{\pi}{1-\pi}}{d}\right)\phi\left(\frac{-c - \ln\frac{\pi}{1-\pi}}{d}\right)$$

$$= \frac{\phi\left(\frac{-c - \ln\frac{\pi}{1-\pi}}{d}\right)}{d\pi(1-\pi)^2}\left(-(1-\pi) - \pi + \frac{1}{d^2}(c + \ln\frac{\pi}{1-\pi})\right)$$

$$= \frac{\phi\left(\frac{-c - \ln\frac{\pi}{1-\pi}}{d}\right)}{d\pi(1-\pi)^2}\left(\frac{1}{d^2}(c + \ln\frac{\pi}{1-\pi}) - 1\right). \tag{46}$$

□

**Lemma 12.** *The second derivative (with respect to $\pi$ while other parameters are constant) of the natural risk of a classifier with parameters $(c, d)$ defined according to Equations* (39) *and* (40) *is given by*

$$\frac{Ce^{-\frac{c^2 + (\ln\frac{\pi}{1-\pi})^2}{2d^2}}}{d(\sqrt{\pi(1-\pi)})^3}\left((\frac{c}{d^2} - 1)\left((\frac{\pi}{1-\pi})^{-\frac{c}{d^2}+\frac{1}{2}} + (\frac{1-\pi}{\pi})^{-\frac{c}{d^2}+\frac{1}{2}}\right) + \frac{\ln\frac{\pi}{1-\pi}}{d^2}\left((\frac{\pi}{1-\pi})^{-\frac{c}{d^2}+\frac{1}{2}} - (\frac{1-\pi}{\pi})^{-\frac{c}{d^2}+\frac{1}{2}}\right)\right) \tag{47}$$

*where $C$ is a positive constant.*

*Proof.* Using Lemma 11, we have

$$\frac{d^2}{d\pi^2}R_{c,d}(\pi) = F''(\pi) + F''(1 - \pi)$$

$$= \frac{\phi\left(\frac{-c - \ln\frac{\pi}{1-\pi}}{d}\right)}{d\pi(1-\pi)^2}\left(\frac{1}{d^2}(c + \ln\frac{\pi}{1-\pi}) - 1\right) + \frac{\phi\left(\frac{-c + \ln\frac{\pi}{1-\pi}}{d}\right)}{d\pi^2(1-\pi)}\left(\frac{1}{d^2}(c - \ln\frac{\pi}{1-\pi}) - 1\right)$$

$$= \frac{Ce^{-\frac{c^2 + (\ln\frac{\pi}{1-\pi})^2}{2d^2}}}{d(\sqrt{\pi(1-\pi)})^3}\left((\frac{c}{d^2} - 1)\left((\frac{\pi}{1-\pi})^{-\frac{c}{d^2}+\frac{1}{2}} + (\frac{1-\pi}{\pi})^{-\frac{c}{d^2}+\frac{1}{2}}\right) + \frac{\ln\frac{\pi}{1-\pi}}{d^2}\left((\frac{\pi}{1-\pi})^{-\frac{c}{d^2}+\frac{1}{2}} - (\frac{1-\pi}{\pi})^{-\frac{c}{d^2}+\frac{1}{2}}\right)\right) \tag{48}$$

□

**Lemma 13.** *Consider the following function*

$$S(x) = \alpha x^2 - \ln\left(x(e^{Ax} - e^{-Ax}) - \beta(A - 0.5)(e^{Ax} + e^{-Ax})\right) \tag{49}$$

*where $A$ and $\beta$ are positive real numbers and $\alpha \geq 0$. There exists $x_0 \in \mathbb{R}^+ \cup \{0\}$ such that the domain of S(x) is $(-\infty, -x_0) \cup (x_0, \infty)$. Furthermore, there exists $x_1 \in \mathbb{R}^+$ such that $S(x)$ is monotonically decreasing on $(x_0, x_1)$ and monotonically increasing on $(x_1, \infty)$.*

*Proof.* Since $S(x)$ is an even function, we only consider the non-negative part of its domain ($x \geq 0$) as everything is symmetric about zero. We have two distinct cases for $A$. $A \leq 0.5$ implies that the domain of $S$ is $\mathbb{R}$. On the other hand, $A > 0.5$ imposes the following condition on $x \geq 0$ to ensure that the logarithmic term remains valid

$$x(e^{Ax} - e^{-Ax}) - \beta(A - 0.5)(e^{Ax} + e^{-Ax}) \geq 0. \iff \frac{e^{Ax} - e^{-Ax}}{e^{Ax} + e^{-Ax}} \geq \frac{\beta(A - 0.5)}{x}. \tag{50}$$

The RHS is strictly decreasing on $(0, \infty)$ and tends to 0 as $x \to \infty$. However, the LHS, which is $\tanh(Ax)$, is strictly increasing on $(0, \infty)$. Hence, there exists $x_0$ such that the above inequality holds for all $x \in (x_0, \infty)$. Also, The domain of $S$ is $(-\infty, -x_0) \cup (x_0, \infty)$. The proof of first statement is complete.

We break down the proof of the second statement to the following three cases.

1. If $\alpha = 0$, we have

$$S'(x) = -\frac{Ax(e^{Ax} + e^{-Ax}) + e^{Ax} - e^{-Ax} - \beta A(A - 0.5)(e^{Ax} - e^{-Ax})}{x(e^{Ax} - e^{-Ax}) - \beta(A - 0.5)(e^{Ax} + e^{-Ax})} \tag{51}$$

The denominator is always positive on the domain of $S(x)$. When $A \leq 0.5$, it is clear that the numerator is also positive. If $A \geq 0.5$, we have

$$x(e^{Ax} - e^{-Ax}) - \beta(A - 0.5)(e^{Ax} + e^{-Ax}) \geq 0$$
$$\Rightarrow Ax(e^{Ax} - e^{-Ax}) - \beta A(A - 0.5)(e^{Ax} + e^{-Ax}) \geq 0$$
$$\Rightarrow 2Axe^{-Ax} + 2\beta A(A - 0.5)e^{-Ax} + Ax(e^{Ax} - e^{-Ax}) - \beta A(A - 0.5)(e^{Ax} + e^{-Ax}) \geq 0$$
$$\Rightarrow Ax(e^{Ax} + e^{-Ax}) + e^{Ax} - e^{-Ax} - \beta A(A - 0.5)(e^{Ax} - e^{-Ax}) \geq 0 \tag{52}$$

Again, the first inequality is correct on the domain of $S$, and the second and third inequalities follow by multiplying by $A$ and adding a positive value respectively. Thus, $S'(x)$ is negative on $(x_0, \infty)$, and $x_1 = \infty$. Therefore, $S$ is strictly decreasing on $(x_0, \infty)$, and doesn't have an increasing stage. The proof for this case is complete.

2. $\alpha > 0$ and $A \geq 0.5$: Another form of S(x) is

$$S(x) = \alpha x^2 + Ax - \ln\left(x(e^{2Ax} - 1) - \beta(A - 0.5)(e^{2Ax} + 1)\right). \tag{53}$$

We will use this alternative form of $S$ in the following arguments. The first derivative of $S$ is

$$S'(x) = 2\alpha x + A - \frac{e^{2Ax}(2Ax + 1 - 2\beta A(A - 0.5)) - 1}{x(e^{2Ax} - 1) - \beta(A - 0.5)(e^{2Ax} + 1)} \tag{54}$$

and after simplification, we have

$$S''(x) = 2\alpha - \frac{-8A\beta(A - 0.5)e^{2Ax} + 4A^2 e^{2Ax}\left((\beta(A - 0.5))^2 - x^2\right) - (e^{2Ax} - 1)^2}{\left(x(e^{2Ax} - 1) - \beta(A - 0.5)(e^{2Ax} + 1)\right)^2}. \tag{55}$$

Furthermore, (50) implies

$$\frac{e^{Ax} - e^{-Ax}}{e^{Ax} + e^{-Ax}} \geq \frac{\beta(A - 0.5)}{x} \Rightarrow x \geq \beta(A - 0.5). \tag{56}$$

Consequently, $(\beta(A - 0.5))^2 \leq x^2$ holds and we can finally conclude that $S''(x) > 0$. Therefore, $S(x)$ is strictly convex on $(x_0, \infty)$, which guarantees the existence of $x_1$ (potentially infinite) as mentioned in the lemma.

3. Assume that $A \leq 0.5$ and $\alpha > 0$. We want to show that $S'(x)$ has at most one root in $x > 0$, and $S'(x) > 0$ as $x \to \infty$. Recall that when $A \leq 0.5$, the domain of $S$ is $\mathbb{R}$. Note that $S'(0)$ is 0. Let

$$L(x) := 2\alpha x + A \qquad T(x) := \frac{e^{2Ax}(2Ax + 1 - 2\beta A(A - 0.5)) - 1}{x(e^{2Ax} - 1) - \beta(A - 0.5)(e^{2Ax} + 1)} \tag{57}$$

Then we have $S'(x) = L(x) - T(x)$. Assume for now that there exists $a$ such that $T(x)$ is strictly concave on $(0, a)$ and strictly decreasing on $(a, \infty)$. Then, since $L$ is a linear function with a positive slope, $L(x) = T(x)$ can have at most one solution on $(0, a]$. If such solution exists, then $T(a) \leq L(a)$ and hence, no more solutions exist beyond $a$. Otherwise, we will either have $T(a) > L(a)$, and exactly one solution will exist beyond $a$, or have $T(a) < L(a)$ and no solution exists on $x > 0$. In conclusion, there can be at most one solution on $(0, \infty)$, and clearly, $T(x) < L(x)$ as $x \to \infty$.

All that is left to prove is the existence of $a$. We will show that $T'(x)$ is strictly decreasing until $a$, and is negative beyond $a$. Similar to the derivative of $S(x)$, we have

$$T'(x) = \frac{-8A\beta(A - 0.5)e^{2Ax} + 4A^2 e^{2Ax}\left((\beta(A - 0.5))^2 - x^2\right) - (e^{2Ax} - 1)^2}{(x(e^{2Ax} - 1) - \beta(A - 0.5)(e^{2Ax} + 1))^2}$$
$$= \frac{-8A\beta(A - 0.5) + 4A^2\left((\beta(A - 0.5))^2 - x^2\right) - (e^{Ax} - e^{-Ax})^2}{(x(e^{Ax} - e^{-Ax}) - \beta(A - 0.5)(e^{Ax} + e^{-Ax}))^2}. \tag{58}$$

Note that

$$T'(x) \geq 0 \iff (\beta(A - 0.5))^2 - 8A\beta(A - 0.5) \geq (2Ax)^2 + (e^{Ax} - e^{-Ax})^2. \tag{59}$$

While the LHS in the above inequality is constant, the RHS is a strictly increasing function of $x$, meaning that there is exactly one $a$ such that $T'(x) < 0$ for all $x > a$, meaning that $T$ is strictly decreasing beyond $a$. Take this $a$. When $x < a$, the nominator is positive and strictly decreasing in $x$, while the denominator is strictly increasing in $x$, since $A \leq 0.5$. As a result, for $x < a$, $T'(x)$ is strictly decreasing in $x$, meaning that $T$ is strictly concave in this area. We have proved the existence of the promised $a$, and the proof of this case, as well as the entire lemma, is complete.

$\square$

**Lemma 14.** *There exists $\pi^*$ such that $R_{c,d}(\pi)$ is monotonically decreasing on $(\pi^*, 1)$, or equivalently, $R'_{c,d}(\pi)$ is negative on this interval.*

*Proof.* we have

$$R'_{c,d}(\pi) = F'(\pi) - F'(1 - \pi) \tag{60}$$

By definition of $F$ and Equation (45), we have

$$\lim_{\pi \to 1} F'(\pi) = \lim_{\pi \to 1} \Phi\left(\frac{-c - \ln\frac{\pi}{1-\pi}}{d}\right) - \lim_{\pi \to 1} \frac{1}{d(1 - \pi)}\phi\left(\frac{-c - \ln\frac{\pi}{1-\pi}}{d}\right) = 0 - 0 = 0 \tag{61}$$

$$\lim_{\pi \to 1} F'(1 - \pi) = \lim_{\pi \to 1} \Phi\left(\frac{-c + \ln\frac{\pi}{1-\pi}}{d}\right) - \lim_{\pi \to 1} \frac{1}{d\pi}\phi\left(\frac{-c + \ln\frac{\pi}{1-\pi}}{d}\right) = 1 - 0 = 1 \tag{62}$$

Hence,

$$\lim_{\pi \to 1} R'_{c,d} = -1 \tag{63}$$

And it shows that there exists a $\pi^*$ such that $R'_{c,d}(\pi)$ is negative on $(\pi^*, 1)$. $\square$

**Lemma 15.** *Let $G(\pi)$, the natural accuracy gap between the optimal adversarial and Bayes optimal classifiers, be defined as*

$$G(\pi) := G_{\mathcal{D}, \ell_{0\text{-}1}} = R_{\mathcal{D}, \ell}^{nat}\left(w^{adv}, \frac{1}{2}\ln\frac{\pi}{1 - \pi}\right) - R_{\mathcal{D}, \ell}^{nat}\left(w^{nat}, \frac{1}{2}\ln\frac{\pi}{1 - \pi}\right) \tag{64}$$

*where $w^{adv}$ and $w^{nat}$ are the weight of the optimal adversarial and Bayes classifiers, respectively. There exists a $\pi^*$ such that $G(\pi)$ is monotonically decreasing on $(\pi^*, 1)$.*

*Proof.* We show that there exists a $\pi^*$ such that $G'(\pi)$ is negative on $(\pi^*, 1)$. Let $c'$ and $d'$ be defined using Equations (39) and (40) when $\epsilon = 0$. Then, we have

$$G'(\pi) = F'_{c,d}(\pi) - F'_{c,d}(1 - \pi) - F'_{c',d'}(\pi) + F'_{c',d'}(1 - \pi) \tag{65}$$

By definitions of $c'$ and $d'$ and Equation (45), we have

$$\frac{1}{(1 - \pi)}\phi\left(\frac{-c' - \ln\frac{\pi}{1-\pi}}{d'}\right) = \frac{1}{\pi}\phi\left(\frac{-c' + \ln\frac{\pi}{1-\pi}}{d'}\right) \tag{66}$$

Hence,

$$F'_{c',d'}(\pi) - F'_{c',d'}(1 - \pi) = \Phi\left(\frac{-c' - \ln\frac{\pi}{1-\pi}}{d'}\right) - \Phi\left(\frac{-c' + \ln\frac{\pi}{1-\pi}}{d'}\right) \tag{67}$$

and

$$G'(\pi) = \Phi\left(\frac{-c - \ln\frac{\pi}{1-\pi}}{d}\right) - \frac{1}{d(1-\pi)}\phi\left(\frac{-c - \ln\frac{\pi}{1-\pi}}{d}\right) - \Phi\left(\frac{-c + \ln\frac{\pi}{1-\pi}}{d}\right)$$
$$+ \frac{1}{d\pi}\phi\left(\frac{-c + \ln\frac{\pi}{1-\pi}}{d}\right) - \Phi\left(\frac{-c' - \ln\frac{\pi}{1-\pi}}{d'}\right) + \Phi\left(\frac{-c' + \ln\frac{\pi}{1-\pi}}{d'}\right) \tag{68}$$

Using definitions of $d$ and $d'$, it is obvious that $d < d'$. Therefore, there exists $\pi^*$ such that for all $\pi \in (\pi^*, 1)$, we have the following inequalities,

$$\frac{-c + \ln\frac{\pi}{1-\pi}}{d} > \frac{-c' + \ln\frac{\pi}{1-\pi}}{d'} > 0, \tag{69}$$

$$\frac{-c - \ln\frac{\pi}{1-\pi}}{d} < \frac{-c' - \ln\frac{\pi}{1-\pi}}{d'} < 0. \tag{70}$$

Since $\Phi$ is concave on $\mathbb{R}^+$, for any $a.b > 0$ we have $\Phi(b) - \Phi(a) \leq \phi(a)(b-a)$. Setting $a = \frac{-c + \ln\frac{\pi}{1-\pi}}{d}$ and $b = \frac{-c' + \ln\frac{\pi}{1-\pi}}{d'}$ would result in,

$$\Phi\left(\frac{-c' + \ln\frac{\pi}{1-\pi}}{d'}\right) - \Phi\left(\frac{-c + \ln\frac{\pi}{1-\pi}}{d}\right) \leq \phi\left(\frac{-c + \ln\frac{\pi}{1-\pi}}{d}\right)\left(\ln\frac{\pi}{1-\pi}(\frac{1}{d'} - \frac{1}{d}) + \frac{c}{d} - \frac{c'}{d'}\right) \tag{71}$$

Combining above inequalities, we have

$$G'(\pi) = \Phi\left(\frac{-c - \ln\frac{\pi}{1-\pi}}{d}\right) - \Phi\left(\frac{-c' - \ln\frac{\pi}{1-\pi}}{d'}\right) + \Phi\left(\frac{-c' + \ln\frac{\pi}{1-\pi}}{d'}\right) - \Phi\left(\frac{-c + \ln\frac{\pi}{1-\pi}}{d}\right)$$
$$+ \frac{1}{d\pi}\phi\left(\frac{-c + \ln\frac{\pi}{1-\pi}}{d}\right) - \frac{1}{d(1-\pi)}\phi\left(\frac{-c + \ln\frac{\pi}{1-\pi}}{d}\right)$$
$$\leq \frac{1}{d\pi}\phi\left(\frac{-c + \ln\frac{\pi}{1-\pi}}{d}\right) + \phi\left(\frac{-c + \ln\frac{\pi}{1-\pi}}{d}\right)\left(\ln\frac{\pi}{1-\pi}(\frac{1}{d'} - \frac{1}{d}) + \frac{c}{d} - \frac{c'}{d'}\right) \tag{72}$$

where we used the fact that for large enough $\pi$ we have $\Phi\left(\frac{-c - \ln\frac{\pi}{1-\pi}}{d}\right) - \Phi\left(\frac{-c' - \ln\frac{\pi}{1-\pi}}{d'}\right) < 0$, and $\phi \geq 0$. It suffices to show that,

$$\frac{1}{d\pi}\phi\left(\frac{-c + \ln\frac{\pi}{1-\pi}}{d}\right) + \phi\left(\frac{-c + \ln\frac{\pi}{1-\pi}}{d}\right)\left(\ln\frac{\pi}{1-\pi}(\frac{1}{d'} - \frac{1}{d}) + \frac{c}{d} - \frac{c'}{d'}\right) \leq 0$$
$$\iff \frac{1}{d\pi} \leq \ln\frac{\pi}{1-\pi}(\frac{1}{d} - \frac{1}{d'}) + \frac{c'}{d'} - \frac{c}{d} \tag{73}$$

When $\pi \to 1$, the LHS in the last inequality tends to $\frac{1}{d}$, while the RHS tends to $\infty$ (since $d < d'$), and is monotonically increasing. Consequently, we set $\pi^*$ large enough such that the last inequality holds, which completes the proof. $\square$

## D.1. Proof of Equation (13)

We begin by proving the following claim: if $R''_{c,d}$ has no roots in $(0.5, 1)$, then $R_{c,d}$ is monotonically decreasing on $(0.5, 1)$, and if $R''_{c,d}$ has exactly one root in $(0.5, 1)$ and $R''_{c,d}(0.5) > 0$, then $R_{c,d}$ is increasing at first and then decreasing, yielding exactly one global maximum in $(0.5, 1)$. The first part of the claim is true since $R'_{c,d}(0.5) = 0$ and $R_{c,d}$ is eventually decreasing, meaning that it must be concave, and subsequently monotonically decreasing on $(0.5, 1)$. For the second part, $R_{c,d}$ will be convex and increasing at first, and concave after that. Since $R_{c,d}$ is eventually decreasing as well, it will also achieve its global maximum during its convex phase.

Assume $\pi$ is a root of $R''_{c,d}$. By using Lemma 12, we have

$$(\frac{c}{d^2} - 1)\left((\frac{\pi}{1-\pi})^{-\frac{c}{d^2}+\frac{1}{2}} + (\frac{1-\pi}{\pi})^{-\frac{c}{d^2}+\frac{1}{2}}\right) + \frac{\ln\frac{\pi}{1-\pi}}{d^2}\left((\frac{\pi}{1-\pi})^{-\frac{c}{d^2}+\frac{1}{2}} - (\frac{1-\pi}{\pi})^{-\frac{c}{d^2}+\frac{1}{2}}\right) = 0 \qquad (74)$$

Let $x = \ln\frac{\pi}{1-\pi}$, $A = \frac{c}{d^2} - \frac{1}{2}$, and $\beta = d^2$. We should have

$$x(e^{Ax} - e^{-Ax}) - \beta(A - 0.5)(e^{Ax} + e^{-Ax}) = 0 \qquad (75)$$

Similar to proof of first part of Lemma 13, when $A > 0.5$ the above equation has only one solution in $x > 0$ and when $A \leq 0.5$, cannot have any roots in $x > 0$. Moreover, one can verify that when $A > 0$ we have $R''_{c,d}(0.5) > 0$ as well. Thus, $A > 0.5$ and $A \leq 0.5$ exactly differentiate between the two regimes.

Finally, note that

$$A > 0.5 \iff \frac{c}{d^2} > 0.5 \iff \langle \mu, \Sigma^{-1}(\mu - \epsilon z^*)\rangle > 2\|\mu - \epsilon z^*\|^2_{\Sigma^{-1}} \qquad (76)$$

## D.2. Proof of Equation (14)

The proof is similar to proof of Equation (13). We want to analyze the second derivation of $G(\pi)$ defined by Equation (64). Recall that we only consider $\pi \geq \frac{1}{2}$. Assume that $\pi$ is root of $G''(\pi)$. Then we have

$$G''(\pi) = R''_{c,d}(\pi) - R''_{c',d'}(\pi) = 0 \iff R''_{c,d}(\pi) = R''_{c',d'}(\pi) \qquad (77)$$

Recall that for the Bayes optimal classifier, we have $\frac{c'}{d'^2} = \frac{1}{2}$ from Lemma 10. Recall from Lemma 12 we have

$$G''(\pi) = R''_{c,d}(\pi) - R''_{c',d'}(\pi)$$
$$= \frac{Ce^{-\frac{c^2 + (\ln\frac{\pi}{1-\pi})^2}{2d^2}}}{d(\sqrt{\pi(1-\pi)})^3}\left((\frac{c}{d^2} - 1)\left((\frac{\pi}{1-\pi})^{-\frac{c}{d^2}+\frac{1}{2}} + (\frac{1-\pi}{\pi})^{-\frac{c}{d^2}+\frac{1}{2}}\right) + \frac{\ln\frac{\pi}{1-\pi}}{d^2}\left((\frac{\pi}{1-\pi})^{-\frac{c}{d^2}+\frac{1}{2}} - (\frac{1-\pi}{\pi})^{-\frac{c}{d^2}+\frac{1}{2}}\right)\right)$$
$$+ \frac{Ce^{-\frac{c'^2 + (\ln\frac{\pi}{1-\pi})^2}{2d'^2}}}{d'(\sqrt{\pi(1-\pi)})^3} \qquad (78)$$

Setting $A = \frac{c}{d^2} - \frac{1}{2}$ and $x = \ln\frac{\pi}{1-\pi}$ implies

$$G''(\pi) = \frac{Ce^{-\frac{c^2 + (\ln\frac{\pi}{1-\pi})^2}{2d^2}}}{d^3(\sqrt{\pi(1-\pi)})^3}\left(d^2(A - 0.5)(e^{-Ax} + e^{Ax}) + x(e^{-Ax} - e^{Ax}) + \frac{d^3}{d'}e^{\frac{1}{2}\left(x^2(\frac{1}{d^2} - \frac{1}{d'^2}) + \frac{c^2}{d^2} - \frac{c'^2}{d'^2}\right)}\right). \qquad (79)$$

Therefore, in order to have $G''(\pi) = 0$ we must have

$$x(e^{Ax} - e^{-Ax}) - d^2(A - 0.5)(e^{-Ax} + e^{Ax}) = \frac{d^3}{d'}e^{\frac{1}{2}\left(x^2(\frac{1}{d^2} - \frac{1}{d'^2}) + \frac{c^2}{d^2} - \frac{c'^2}{d'^2}\right)}$$
$$\iff \ln\left(x(e^{Ax} - e^{-Ax}) - d^2(A - 0.5)(e^{-Ax} + e^{Ax})\right) = \ln\frac{d^3}{d'} + \frac{1}{2}\left(x^2(\frac{1}{d^2} - \frac{1}{d'^2}) + \frac{c^2}{d^2} - \frac{c'^2}{d'^2}\right). \qquad (80)$$

Let $\beta = d^2$ and $\alpha = \frac{1}{2}(\frac{1}{d^2} - \frac{1}{d'^2})$. It is clear that $\alpha, \beta > 0$. As a result, we have

$$\ln\left(x(e^{Ax} - e^{-Ax}) - \beta(A - 0.5)(e^{-Ax} + e^{Ax})\right) = \ln\frac{d^3}{d'} + \frac{1}{2}\left(x^2(\frac{1}{d^2} - \frac{1}{d'^2}) + \frac{c^2}{d^2} - \frac{c'^2}{d'^2}\right)$$
$$\iff T + \alpha x^2 - \ln\left(x(e^{Ax} - e^{-Ax}) - \beta(A - 0.5)(e^{-Ax} + e^{Ax})\right) = 0$$
$$\iff T + S(x) = 0 \tag{81}$$

where T is a constant. From Lemma 13 we know that $S(x)$ in $(0, \infty)$ has two stages, decreasing at first, and then increasing. Similarly, $T + S(x)$ has these two stage in $x > 0$. We have two cases

1. If $G''(\frac{1}{2}) \leq 0$ or equivalently, using the definition of $G''(\pi)$,

$$2(\frac{c}{d^2} - 1)\frac{e^{-\frac{1}{2}(\frac{c}{d})^2}}{d} \leq -\frac{e^{-\frac{\|\mu\|^2_{\Sigma^{-1}}}{2}}}{2\|\mu\|_{\Sigma^{-1}}} \tag{82}$$

then, one can observe from Equation (79) that $A \leq 0.5$ and per Lemma 13 the domain of $S(x)$ is $(0, \infty)$. Furthermore, $S(0) + T \leq 0$ holds according to $G''(\frac{1}{2}) \leq 0$. From Lemma 13, we know that there exists $x_1$ such that $S(x)$ is decreasing on $(0, x_1)$ and increasing on $(x_1, \infty)$. Combined with the fact that $S(0) + T \leq 0$, we can conclude that $S(x) + T$ has at most one root in $(0, \infty)$. Similarly, $G''$ has at most a root in $(0.5, 1)$. Furthermore, Lemma 15 ensures that $G$ is eventually decreasing. Following the first claim of the proof of Equation (13), we know that $G''(0) \leq 0$ and $G''$ having no roots in $(0.5, 1)$ imply that $G$ is monotonically decreasing on $(0.5, 1)$, which completes the proof in this case.

2. If $G''(\frac{1}{2}) > 0$, or equivalently

$$2(\frac{c}{d^2} - 1)\frac{e^{-\frac{1}{2}(\frac{c}{d})^2}}{d} > -\frac{e^{-\frac{\|\mu\|^2_{\Sigma^{-1}}}{2}}}{2\|\mu\|_{\Sigma^{-1}}} \tag{83}$$

then Lemma 13 shows that $S(x) + T$ has at most two roots in $(0, \infty)$, which implies that $G''$ has at most two roots in $(0.5, 1)$, since it is impossible for $G''$ to have roots corresponding to $x$ out of the domain of $S(x)$. Moreover, $\frac{1}{2}$ is a local minimal of $G$ since $G'(\frac{1}{2}) = 0$ and $G''(\frac{1}{2}) > 0$. As $G$ is eventually decreasing, $G$ cannot remain convex on the entirety of $(0.5, 1)$, meaning that $G''$ has either exactly one or two roots. When $G''$ has one root, similar to the proof of Equation (13), $G$ is increasing at first and then decreasing on $(0.5, 1)$, with exactly one global maximum. When $G''$ has two roots, $G$ is convex and increasing at first, then concave, and then convex again. Note that since $G$ is eventually decreasing, it must be decreasing on the entirety of its final convex phase, which means that it has already become decreasing on its concave phase, and has reached its maximum value in this phase as well. Once again, we have shown that $G$ is increasing at first and then decreasing on $(0.5, 1)$, and the proof of this case, as well as the entire theorem, is complete.

## E. Proof of Theorem 5

Based on Proposition 3, we can write $w^{adv} = \Sigma^{-1}(\mu - \epsilon z^*)$, where $z^* = \arg\min_{\|z\|_\infty \leq 1}\|\mu - \epsilon z\|^2_{\Sigma^{-1}}$. In order for the two classifiers to be equivalent, we need $w^{adv} = \lambda w^{nat}$ for some $\lambda > 0$. Since $w^{nat} = \Sigma^{-1}\mu$, this would imply $z^* = \frac{\mu(1-\lambda)}{\epsilon}$. Moreover, recall from Appendix C that whenever $w_i^{adv} \neq 0$ we have $z_i^* = \text{sign}(w_i^{adv})$, which here it would imply $z_i^* = \text{sign}((\Sigma^{-1}\mu)_i)$ whenever $(\Sigma^{-1}\mu)_i \neq 0$. Besides, based on $\lambda > 1$ or $\lambda < 1$, we must either have $\text{sign}(z_i^*) = \text{sign}(\mu_i)$ or $\text{sign}(z_i^*) = -\text{sign}(\mu_i)$ for all $i$. Since $\Sigma$ is positive definite, we must have $\langle \mu, \Sigma^{-1}\mu \rangle \geq 0$, therefore we always have $\text{sign}(z_i^*) = \text{sign}(\mu_i)$, i.e. $\lambda < 1$. In conclusion, we have

$$(\Sigma^{-1}\mu)_i \neq 0 \implies \frac{\mu_i(1-\lambda)}{\epsilon} = \text{sign}((\Sigma^{-1}\mu)) \implies \mu_i = c\,\text{sign}((\Sigma^{-1}\mu)_i) \tag{84}$$
$$(\Sigma^{-1}\mu)_i = 0 \implies |z_i^*| \leq 1 \implies |\mu_i| \leq c \tag{85}$$

where $c = \frac{\epsilon}{1-\lambda} > \epsilon$. Conversely, note that whenever the above conditions hold, one can easily verify that $w^{adv} = \Sigma^{-1}\mu$ is a valid solution, completing the proof. $\square$

# F. Proof of Theorem 6

Recall that $w^{adv} = \Sigma^{-1}(\mu - \epsilon z^*)$, where $z^* = \arg\min_{\|z\|_\infty \leq 1}\|\mu - \epsilon z\|_{\Sigma^{-1}}^2$. Also, when $\pi_+ = \pi_- = \frac{1}{2}$ we have

$$R_{\mathcal{D},\ell}^{nat}(w^{adv}) = \Phi\left(\frac{-\langle w^{adv}, \mu\rangle}{\|w^{adv}\|_\Sigma}\right) = \Phi\left(\frac{-\|\mu\|_{\Sigma^{-1}}^2 + \epsilon\langle z^*, \Sigma^{-1}\mu\rangle}{\|\mu - \epsilon z^*\|_{\Sigma^{-1}}}\right). \tag{86}$$

Furthermore, using the Hölder's inequality we have $\langle z^*, \Sigma^{-1}\mu\rangle \leq \|z^*\|_\infty\|\Sigma^{-1}\mu\|_1 \leq \|\Sigma^{-1}\mu\|_1$, and by definition of $z^*$ we have $\|\mu - \epsilon\,\mathrm{sign}(\Sigma^{-1}\mu)\|_{\Sigma^{-1}} \geq \|\mu - \epsilon z^*\|_{\Sigma^{-1}}$. Taking $\epsilon < \frac{\|\mu\|_{\Sigma^{-1}}^2}{2\|\Sigma^{-1}\mu\|_1}$ into account, we have

$$R_{\mathcal{D},\ell}^{nat}(w^{adv}) \leq \Phi\left(\frac{-\|\mu\|_{\Sigma^{-1}}^2 + \epsilon\|\Sigma^{-1}\mu\|_1}{\sqrt{\|\mu\|_{\Sigma^{-1}}^2 - 2\epsilon\|\Sigma^{-1}\mu\|_1 + \epsilon^2\|\mathrm{sign}(\Sigma^{-1}\mu)\|_{\Sigma^{-1}}^2}}\right). \tag{87}$$

For simplicity, we define $A := \frac{\|\Sigma^{-1}\mu\|_1}{\|\mu\|_{\Sigma^{-1}}^2}$ and $B := \frac{\|\mathrm{sign}(\Sigma^{-1}\mu)\|_{\Sigma^{-1}}^2}{\|\mu\|_{\Sigma^{-1}}^2}$. Then

$$\begin{aligned}
R_{\mathcal{D},\ell}^{nat}(w^{adv}) &\leq \Phi\left(-\|\mu\|_{\Sigma^{-1}}\frac{1 - \epsilon A}{\sqrt{1 - 2\epsilon A + \epsilon^2 B}}\right) = \Phi\left(-\|\mu\|_{\Sigma^{-1}}\frac{1}{\sqrt{1 + \frac{\epsilon^2(B - A^2)}{(1 - \epsilon A)^2}}}\right) \\
&\leq \Phi\left(-\|\mu\|_{\Sigma^{-1}}\frac{1}{\sqrt{1 + 4\epsilon^2(B - A^2)}}\right) \\
&\leq \Phi\left(-\|\mu\|_{\Sigma^{-1}}(1 - 2\epsilon^2(B - A^2))\right) \tag{88}
\end{aligned}$$

where the second and third inequalities follow from the facts that $\epsilon \leq \frac{1}{2A}$ and $\frac{1}{\sqrt{1+x}} \geq 1 - \frac{x}{2}$ for all $x > -1$. Recall that $C_{\Sigma,\mu} = \|\mathrm{sign}(\Sigma^{-1}\mu)\|_{\Sigma^{-1}}^2 - \frac{\|\Sigma^{-1}\mu\|_1^2}{\|\mu\|_{\Sigma^{-1}}^2}$. Thus, it is straightforward to see

$$g \leq \Phi(-\|\mu\|_{\Sigma^{-1}} + \frac{2C_{\Sigma,\mu}\epsilon^2}{\|\mu\|_{\Sigma^{-1}}}) - \Phi(-\|\mu\|_{\Sigma^{-1}}). \tag{89}$$

Besides, $\Phi$ is convex on $\mathbb{R}^-$. Therefore, for any $a, b \leq 0$ we have $\Phi(b) - \Phi(a) \leq \frac{e^{\frac{-b^2}{2}}}{\sqrt{2\pi}}(b - a)$. Consequently, we have

$$g \leq \frac{2e^{\frac{\left(-\|\mu\|_{\Sigma^{-1}} + \frac{2C_{\Sigma,\mu}\epsilon^2}{\|\mu\|_{\Sigma^{-1}}}\right)^2}{2}}C_{\Sigma,\mu}\epsilon^2}{\sqrt{2\pi}\|\mu\|_{\Sigma^{-1}}}. \tag{90}$$

Moreover, with $\epsilon \leq \frac{\|\mu\|_{\Sigma^{-1}}}{2\sqrt{C_{\Sigma,\mu}}}$ we have $-\|\mu\|_{\Sigma^{-1}} + \frac{2C_{\Sigma,\mu}\epsilon^2}{\|\mu\|_{\Sigma^{-1}}} \leq \frac{-\|\mu\|_{\Sigma^{-1}}}{2}$ and

$$g \leq \frac{2e^{\frac{-\|\mu\|_{\Sigma^{-1}}^2}{8}}C_{\Sigma,\mu}\epsilon^2}{\sqrt{2\pi}\|\mu\|_{\Sigma^{-1}}}. \tag{91}$$

In order to prove the lower bound, we first redefine $A$ and $B$ as $A := \frac{\langle z^*, \Sigma^{-1}\mu\rangle}{\|\mu\|_{\Sigma^{-1}}^2}$ and $B := \frac{\|z^*\|_{\Sigma^{-1}}^2}{\|\mu\|_{\Sigma^{-1}}^2}$. Now we have

$$\begin{aligned}
R_{\mathcal{D},\ell}^{nat}(w^{adv}) &= \Phi\left(-\|\mu\|_{\Sigma^{-1}}\frac{1 - \epsilon A}{\sqrt{1 - 2\epsilon A + \epsilon^2 B}}\right) \\
&= \Phi\left(\frac{-\|\mu\|_{\Sigma^{-1}}}{\sqrt{1 + \frac{\epsilon^2(B - A^2)}{(1 - \epsilon A)^2}}}\right) \\
&\geq \Phi\left(\frac{-\|\mu\|_{\Sigma^{-1}}}{\sqrt{1 + \epsilon^2(B - A^2)}}\right). \tag{92}
\end{aligned}$$

Furthermore, by the definition of $z^*$, when $\Sigma$ is diagonal and $\min_{i \in [d]} |\mu_i| \geq \epsilon$, it is easy to observe that $z^* = \text{sign}(\mu) = \text{sign}(\Sigma^{-1}\mu)$. As a result, we have $B - A^2 = \frac{C_{\Sigma,\mu}}{\|\mu\|^2_{\Sigma^{-1}}}$. In addition, the conditions of the theorem stipulate that $\frac{C_{\Sigma,\mu}\epsilon^2}{\|\mu\|^2_{\Sigma^{-1}}} \leq \frac{1}{4}$, and for $x \leq \frac{1}{2}$ we have $\frac{1}{\sqrt{1+x}} \leq 1 - \frac{x}{3}$. Combined with (92), we have

$$R^{nat}_{\mathcal{D},\ell}(w^{adv}) \geq \Phi\left(-\|\mu\|_{\Sigma^{-1}}\left(1 - \frac{C_{\Sigma,\mu}\epsilon^2}{3\|\mu\|^2_{\Sigma^{-1}}}\right)\right) \implies g \geq \Phi(-\|\mu\|_{\Sigma^{-1}} + \frac{C_{\Sigma,\mu}\epsilon^2}{3\|\mu\|_{\Sigma^{-1}}}) - \Phi(-\|\mu\|_{\Sigma^{-1}}). \quad (93)$$

Again, the convexity of $\Phi$ on $\mathbb{R}^-$ yields that for any $a, b \leq 0$ we have $\Phi(b) - \Phi(a) \geq \frac{e^{\frac{-a^2}{2}}}{\sqrt{2\pi}}(b - a)$. Thus, we have

$$g \geq \frac{e^{\frac{-\|\mu\|^2_{\Sigma^{-1}}}{2}} C_{\Sigma,\mu}\epsilon^2}{3\sqrt{2\pi}\|\mu\|_{\Sigma^{-1}}}. \quad (94)$$

$\square$

# G. Numerical Results

In the section, we present numerical experiments that validate Theorems 4 and 6. Furthermore, we discover a new trend in the gap as the adversarial budget increases. A theoretical analysis of this trend can be an interesting line of future work.

## G.1. Class Imbalance

In Section 4.1 we analyzed the impact of class imbalance on the natural risk of the optimal adversarial classifier and the natural risk gap between the two classifiers. Theorem 4 proposes the necessary and sufficient condition for transitioning between two different regimes, one with a single global maximum and another with two global maxima. Here, we validate this theorem in two cases. In the first case, we set $\mu = (1.5, 2, 4)$ and $\Sigma = 3I_3$. In Figure 2, we can observe the occurrence of two distinct regimes for both the natural risk of the optimal adversarial classifier and the gap, which demonstrates our desired result. For example, $\epsilon = 1.5$ does not satisfy the condition of Equation (13). Hence, the natural risk of adversarial classifier has a concave shape with a single maximum at $\frac{1}{2}$. On the other hand, $\epsilon = 2.5$ satisfies this condition. As a result, we have the promised two maxima located symmetrically around $\frac{1}{2}$ and a local minimum at $\frac{1}{2}$.

Similarly, for a non-diagonal covariance matrix, we consider $\mu = (2, 1, 3)$ and

$$\Sigma = \begin{pmatrix} 2 & 1 & 1 \\ 1 & 2 & 1.5 \\ 1 & 1.5 & 3 \end{pmatrix}.$$

Figure 3 shows the result of this case, which conforms with our theoretical results. Note that the natural risk of the Bayes classifier does not depend on the adversarial budget ($\epsilon$), and as expected, we only have a single maximum at $\frac{1}{2}$ for this risk.

In this section and the next ones, in order to obtain exact risks for the optimal adversarial classifier and the gap, we need to numerically solve Equation (10) and obtain $z^*$. We used CVXPY (Diamond & Boyd, 2016) to this end.

## G.2. Gap Lower and Upper Bounds

In this experiment, we validate our bounds for the natural risk gap between the two classifiers. Theorem 6 provides a general upper bound for all balanced binary Gaussian mixture settings, as well as a lower bound when the covariance matrix is diagonal. For the upper bound, we consider Equation (90) as it provides a tighter bound. Firstly, we evaluate both the upper and lower bounds for a diagonal covariance matrix. With $\mu = (1.5, 2, 4)$ and $\Sigma = 3I_3$, Figure 4a depicts the exact value of the gap as a function of $\epsilon$ along with our proposed lower and upper bounds. Since for the non-diagonal setting we only have an upper bound, it is the only bound plotted in Figure 4b, when $\mu = (1, 1, 1.5)$ and

$$\Sigma = \begin{pmatrix} 2 & 0.5 & 1 \\ 0.5 & 2 & 1.5 \\ 1 & 1.5 & 4 \end{pmatrix}.$$

While our bounds are tight enough for the cases studied here, we show that they can be improved in the next section.

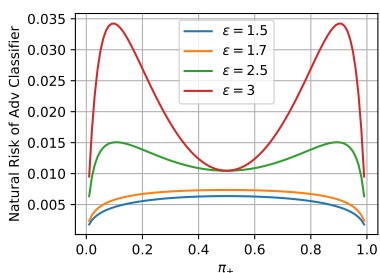

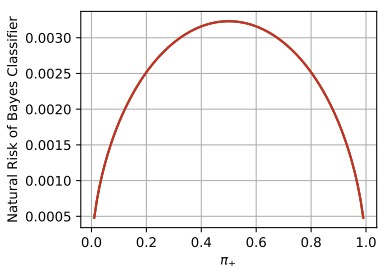

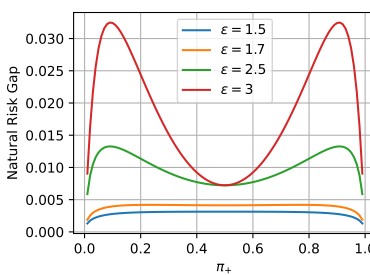

(a) The natural risk of the optimal adversarial classifier.

(b) The natural risk of the Bayes classifier.

(c) The natural risk gap between the optimal adversarial and the Bayes classifier.

*Figure 2.* Natural risk curves obtained with parameters $\mu = (1.5, 2, 4)$ and $\Sigma = 3I_3$.

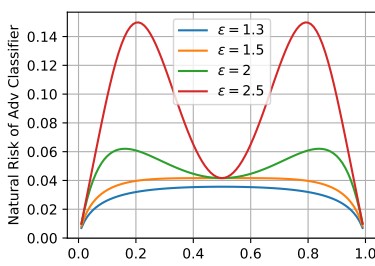

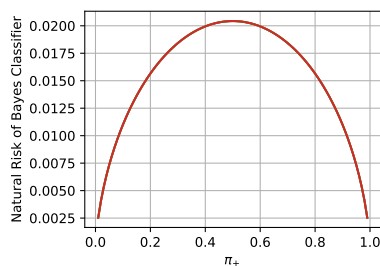

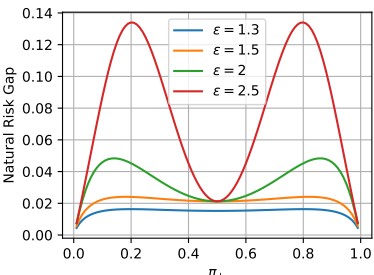

(a) The natural risk of the optimal adversarial classifier.

(b) The natural risk of the Bayes classifier.

(c) The natural risk gap between the optimal adversarial and the Bayes classifier.

*Figure 3.* Natural risk curves obtained with parameters $\mu = (2, 1, 3)$ and $\Sigma = \begin{pmatrix} 2 & 1 & 1 \\ 1 & 2 & 1.5 \\ 1 & 1.5 & 3 \end{pmatrix}$.

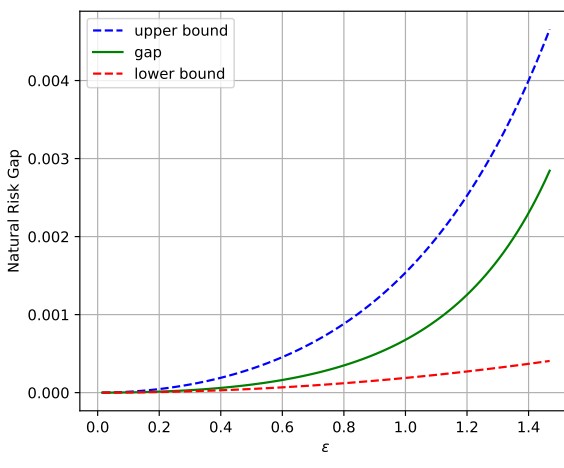

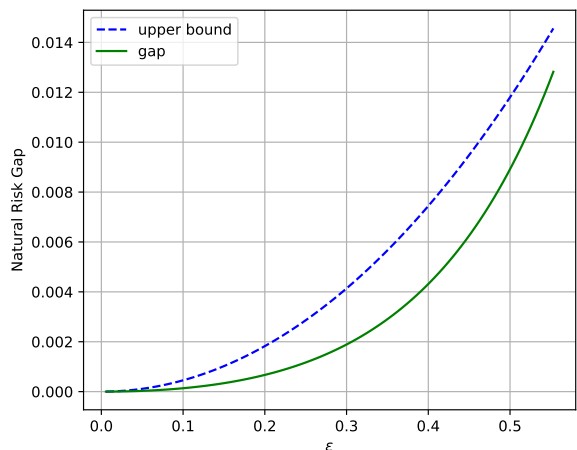

(a) Chosen fixed parameters are $\mu = (1.5, 2, 4)$ and $\Sigma = 3I_3$.

(b) Chosen fixed parameters are $\mu = (1, 1, 1.5)$ and $\Sigma = \begin{pmatrix} 2 & 0.5 & 1 \\ 0.5 & 2 & 1.5 \\ 1 & 1.5 & 4 \end{pmatrix}$. We only have an upper bound in this case.

*Figure 4.* The natural risk gap as a function of $\epsilon$, along with the proposed upper and lower bounds.

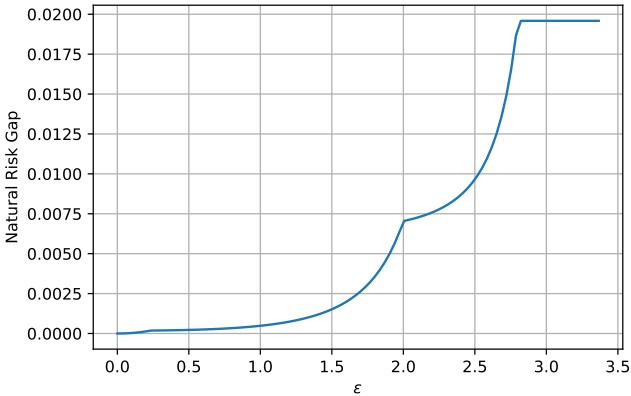

*Figure 5.* The natural risk gap as a function of $\epsilon$ with fixed parameters $\mu = (1, 2, 3, 3.4)$ and $\Sigma = \begin{pmatrix} 3 & 1 & 1 & 0 \\ 1 & 3 & 0 & 0 \\ 1 & 0 & 3 & 1 \\ 0 & 0 & 1 & 3 \end{pmatrix}$. This figure indicates

the existence of break points which change the trend of the curve.

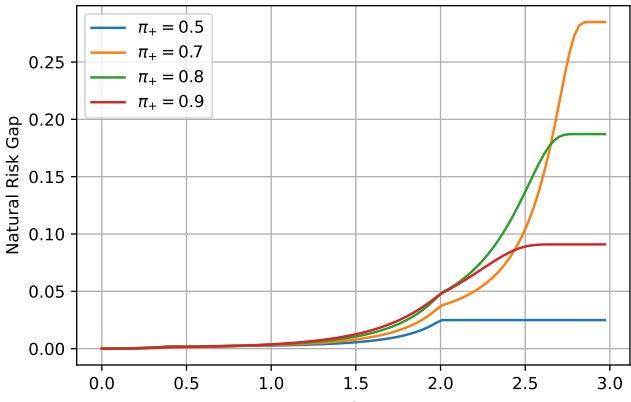

*Figure 6.* The gap as a function of $\epsilon$ for different $\pi_+$ with fixed parameters $\mu = (1, 2, 3)$ and $\Sigma = \begin{pmatrix} 3 & 1 & 1 \\ 1 & 3 & 0 \\ 1 & 0 & 3 \end{pmatrix}$.

### G.3. The Gap and Adversarial Budget

In this experiment, we aim to delve deeper into the behavior of the natural risk gap as a function of the adversarial budget, $\epsilon$. We plot the gap as a function of $\epsilon$ with $\pi_+ = \frac{1}{2}$ and $\mu = (1, 2, 3, 3.4)$ and

$$\Sigma = \begin{pmatrix} 3 & 1 & 1 & 0 \\ 1 & 3 & 0 & 0 \\ 1 & 0 & 3 & 1 \\ 0 & 0 & 1 & 3 \end{pmatrix}.$$

As shown in Figure 5, the curve has 3 break points, i.e. points where the function is non-differentiable. The first is between 0 and 0.5, the second is 2 and the last occurs between 2.5 and 3. The existence of these points is due to the $\ell_\infty$ norm constraint of the optimization problem in Equation (10). The piecewise behavior of the gap encourages locating the break points and providing a tighter gap between each two break points in the future.

## G.4. The Gap and Imbalanced Priors

In this experiment, we compare the trend of the natural risk gap as a function of $\epsilon$, obtained with different $\pi_+$, when we have fixed $\mu = (1, 2, 3)$ and

$$\Sigma = \begin{pmatrix} 3 & 1 & 1 \\ 1 & 3 & 0 \\ 1 & 0 & 3 \end{pmatrix}.$$

We have two observations from Figure 6, which their theoretical study is left as future work.

1. In contrast to the case of $\pi_+ = \frac{1}{2}$ where the gap remains constant beyond $\epsilon = 2$, when $\pi_+ \neq \frac{1}{2}$ the gap still increases after this point. This observation shows that the convergence point of the gap, i.e. the $\epsilon$ beyond which the gap is constant, varies with $\pi_+$.

2. The worst priors for the gap vary with $\epsilon$. For instance, when $\epsilon \in (1.5, 2)$, $\pi_+ = 0.8$ achieves a higher gap compared to $\pi_+ = 0.7$, which reverses beyond $\epsilon > 2.5$.