# OpenReview forum: "The Interplay between Distribution Parameters and the Accuracy-Robustness Tradeoff in Classification"
_ICML.cc/2021/Workshop/AML — ICML 2021 Workshop AML Poster_

### Official Review · Reviewer_uBmV · 2021-06-20
**More interpretation of the insight of the derived formula is needed**

**Rating:** Accept
**Confidence:** 2

**Review:**

In this paper, a detailed and solid mathematical analysis of the accuracy reduction of the standard model in the binary Gaussian mixture classification task after admittedly training is carried out. The accuracy difference between the optimal Bayesian classifier and the adversarial classifier is deduced. Some important components, including the influence of Gaussian distribution parameters on the accuracy gap, are also studied. This makes a good contribution to the theoretical study of robustness and makes up for some phenomena that cannot be revealed in the empirical experimental data.
However, in the derivation of mathematical formulas, the paper lays too much emphasis on boundary conditions and puts a lot of proof parts in the appendix. However, the lack of interpretation of the insight of the derived formula makes it difficult to understand the purpose of many formula derivations when reading the article. The writing of the article may need to be optimized.

---

### Decision · Program_Chairs · 2021-06-21

**Decision:**

Accept (Poster)

**Comment:**

This paper provided a detailed and solid mathematical analysis of the accuracy reduction of the standard model in the binary Gaussian mixture classification task after admittedly training is carried out. The paper can be improved by considering the reviewer's comments.